# Leveraging Factored Action Spaces for Efficient Offline Reinforcement Learning in Healthcare

**Shengpu Tang**[1]   **Maggie Makar**[1]   **Michael W. Sjoding**[2]   **Finale Doshi-Velez**[3]   **Jenna Wiens**[1]

[1]Division of Computer Science & Engineering, University of Michigan, Ann Arbor, MI, USA
[2]Division of Pulmonary and Critical Care, Michigan Medicine, Ann Arbor, MI, USA
[3]SEAS, Harvard University, Cambridge, MA, USA

Correspondence to: {tangsp,wiensj}@umich.edu
Reviewed on OpenReview: https://openreview.net/forum?id=Jd70afzIvJ4

## Abstract

Many reinforcement learning (RL) applications have combinatorial action spaces, where each action is a composition of sub-actions. A standard RL approach ignores this inherent factorization structure, resulting in a potential failure to make meaningful inferences about rarely observed sub-action combinations; this is particularly problematic for offline settings, where data may be limited. In this work, we propose a form of linear Q-function decomposition induced by factored action spaces. We study the theoretical properties of our approach, identifying scenarios where it is guaranteed to lead to zero bias when used to approximate the Q-function. Outside the regimes with theoretical guarantees, we show that our approach can still be useful because it leads to better sample efficiency without necessarily sacrificing policy optimality, allowing us to achieve a better bias-variance trade-off. Across several offline RL problems using simulators and real-world datasets motivated by healthcare, we demonstrate that incorporating factored action spaces into value-based RL can result in better-performing policies. Our approach can help an agent make more accurate inferences within underexplored regions of the state-action space when applying RL to observational datasets.

## 1 Introduction

In many real-world decision-making problems, the action space exhibits an inherent combinatorial structure. For example, in healthcare, an action may correspond to a combination of drugs and treatments. When applying reinforcement learning (RL) to these tasks, past work [1–4] typically considers each combination a distinct action, resulting in an exponentially large action space (Figure 1a). This is inefficient as it fails to leverage any potential independence among dimensions of the action space.

This type of factorization structure in action space could be incorporated when designing the architecture of function approximators for RL (Figure 1b). Similar ideas have been used in the past, primarily to improve online exploration [5, 6], or to handle multiple agents [7–11] or multiple rewards [12]. However, the applicability of this approach has not been systematically studied, especially in offline settings and when the MDP presents no additional structure (e.g., when the state space cannot be explicitly factorized).

In this work, we develop an approach for offline RL with factored action spaces by learning linearly decomposable Q-functions. First, we study the theoretical properties of this approach, investigating the sufficient and necessary conditions for it to lead to an unbiased estimate of the Q-function (i.e., zero approximation error). Even when the linear decomposition is biased, we note that our approach

36th Conference on Neural Information Processing Systems (NeurIPS 2022).

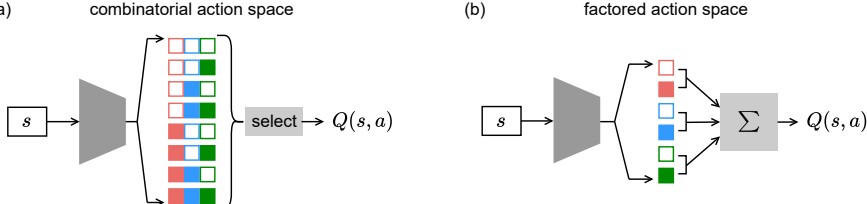

Figure 1: Illustration of Q-network architectures, which take the state $s$ as input and output $Q(s, a)$ for a selected action. In this example, the action space $\mathcal{A}$ consists of $D = 3$ binary sub-action spaces $\{\square, \blacksquare\}$, $\{\square, \blacksquare\}$ and $\{\square, \blacksquare\}$. (a) Learning with the combinatorial action space requires $2^3 = 8$ output heads (exponential in $D$), one for each combination of sub-actions. (b) Incorporating the linear Q decomposition for the factored action space requires $2 \times 3 = 6$ output heads (linear in $D$).

leads to a reduction of variance, which in turn leads to an improvement in sample efficiency. Lastly, we show that when sub-actions exhibit certain structures (e.g., when two sub-actions "reinforce" their independent effects), the linear approximation, though biased, can still lead to the optimal policy. We test our approach in offline RL domains using a simulator [13] and a real clinical dataset [2], where domain knowledge about the relationship among actions suggests our proposed factorization approach is applicable. Empirically, our approach outperforms a non-factored baseline when the sample size is limited, even when the theoretical assumptions (around the validity of a linear decomposition) are not perfectly satisfied. Qualitatively, in the real-data experiment, our approach learns policies that better capture the effect of less frequently observed treatment combinations.

Our work provides both theoretical insights and empirical evidence for RL practitioners to consider this simple linear decomposition for value-based RL approaches. Our contribution complements many popular offline RL methods focused on distribution shift (e.g., BCQ [14]) and goes beyond pessimism-only methods by leveraging domain knowledge. Compatible with any algorithm that has a Q-function component, we expect our approach will lead to gains for offline RL problems with combinatorial action spaces where data are limited and when domain knowledge can be used to check the validity of theoretical assumptions.

## 2 Problem Setup

We consider Markov decision processes (MDPs) defined by a tuple $\mathcal{M} = (\mathcal{S}, \mathcal{A}, p, r, \mu_0, \gamma)$, where $\mathcal{S}$ and $\mathcal{A}$ are the state and action spaces, $p(s'|s, a)$ and $r(s, a)$ are the transition and instantaneous reward functions, $\mu_0(s)$ is the initial state distribution, and $\gamma \in [0, 1]$ is the discount factor. A probabilistic policy $\pi(a|s)$ specifies a mapping from each state to a probability distribution over actions. For a deterministic policy, $\pi(s)$ refers to the action with $\pi(a|s) = 1$. The state-value function is defined as $V^\pi(s) = \mathbb{E}_\pi \mathbb{E}_{\mathcal{M}} \left[ \sum_{t=1}^\infty \gamma^{t-1} r_t \mid s_1 = s \right]$. The action-value function, $Q^\pi(s, a)$, is defined by further restricting the action taken from the starting state. The goal of RL is to find a policy $\pi^* = \arg\max_\pi \mathbb{E}_{s \sim \mu_0}[V^\pi(s)]$ (or an approximation) that has the maximum expected performance.

### 2.1 Factored Action Spaces

While the standard MDP definition abstracts away the underlying structure within the action space $\mathcal{A}$, in this paper, we explicitly express a factored action space as a Cartesian product of $D$ sub-action spaces, $\mathcal{A} = \bigotimes_{d=1}^{D} \mathcal{A}_d = \mathcal{A}_1 \times \cdots \times \mathcal{A}_D$. We use $\boldsymbol{a} \in \mathcal{A}$ to denote each action, which can be written as a vector of sub-actions $\boldsymbol{a} = [a_1, \ldots, a_D]$, with each $a_d \in \mathcal{A}_d$. In general, a sub-action space can be discrete or continuous, and the cardinalities of discrete sub-action spaces are not required to be the same. For clarity of analysis and illustration, we consider discrete sub-action spaces in this paper.

### 2.2 Linear Decomposition of Q Function

The traditional factored MDP literature almost exclusively considers state space factorization [15]. In contrast, here we capitalize on action space factorization to parameterize value functions. Specifically, our approach considers a linear decomposition of the $Q$ function, as illustrated in Figure 1b:

$$Q^\pi(s, \boldsymbol{a}) = \sum_{d=1}^{D} q_d(s, a_d). \tag{1}$$

Each component $q_d(s, a_d)$ in the summation is allowed to condition on the full state space $s$ and only one sub-action $a_d$. While similar forms of decomposition have been used in past work, there are key

differences in how the summation components are parameterized. In the multi-agent RL literature, each component $q_d(s_d, a_d)$ can only condition on the corresponding state space of the $d$-th agent [e.g., 8, 9]. The decomposition in Eqn. (1) also differs from a related form of decomposition considered by Juozapaitis et al. [12] where each component $q_d(s, \boldsymbol{a})$ can condition on the full action $\boldsymbol{a}$. To the best of our knowledge, we are the first to consider this specific form of Q-function decomposition backed by both theoretical rigor and empirical evidence; in addition, we are the first to apply this idea to offline RL. We discuss other related work in Section 5.

## 3 Theoretical Analyses

In this section, we study the theoretical properties of the linear Q-function decomposition induced by factored action spaces. We first present sufficient and necessary conditions for our approach to yield unbiased estimates, and then analyze settings in which our approach can reduce variance without sacrificing policy performance when the conditions are violated. Finally, we discuss how domain knowledge may be used to check the validity of these conditions, providing examples in healthcare.

### 3.1 Sufficient Conditions for Zero Bias

If we consider the total return of $D$ MDPs running in parallel, where each MDP is defined by their respective state space $\mathcal{S}_d$ and action space $\mathcal{A}_d$, then the desired linear decomposition holds for the MDP defined by the joint state space $\bigotimes_{d=1}^{D} \mathcal{S}_d$ and joint action space $\bigotimes_{d=1}^{D} \mathcal{A}_d$ (formally discussed in Appendix B.1). However, this relies on an explicit, known state space factorization, limiting its applicability. In contrast, we now present a generalization that forgoes the explicit factorization of the state space by making use of state abstractions. Intuitively, the MDP should have some implicit factorization, such that it is homomorphic to $D$ parallel MDPs. It is, however, not a requirement that this factorization is known, as long as it exists.

**Theorem 1.** *Given an MDP defined by $\mathcal{S}, \mathcal{A}, p, r$ and a policy $\pi : \mathcal{S} \to \Delta(\mathcal{A})$, where $\mathcal{A} = \bigotimes_{d=1}^{D} \mathcal{A}_d$ is a factored action space with $D$ sub-action spaces, if there exists $D$ unique corresponding state abstractions $\phi = [\phi_1, \cdots, \phi_D]$ where $\phi_d : \mathcal{S} \to \mathcal{Z}_d$, $z_d = \phi_d(s)$, $z'_d = \phi_d(s')$, such that for all $s, a, s'$ the following holds:*

$$\sum_{\tilde{s} \in \phi^{-1}(\phi(s'))} p(\tilde{s}|s, \boldsymbol{a}) = \prod_{d=1}^{D} p_d(z'_d | z_d, a_d) \tag{2}$$

$$r(s, \boldsymbol{a}) = \sum_{d=1}^{D} r_d(z_d, a_d) \tag{3} \qquad\qquad \pi(\boldsymbol{a}|s) = \prod_{d=1}^{D} \pi_d(a_d | z_d) \tag{4}$$

*for some $p_d : \mathcal{Z}_d \times \mathcal{A}_d \to \Delta(\mathcal{Z}_d)$, $r_d : \mathcal{Z}_d \times \mathcal{A}_d \to \mathbb{R}$, and $\pi_d : \mathcal{Z}_d \to \Delta(\mathcal{A}_d)$, then the Q-function of policy $\pi$ can be expressed as $Q^\pi(s, \boldsymbol{a}) = \sum_{d=1}^{D} q_d(s, a_d)$.*

In Appendix B.2, we present an induction-based proof of Theorem 1. Since every assumption is used in key steps of the proof, we conjecture that the sufficient conditions cannot be relaxed in general. Consequently, if the sufficient conditions are satisfied, then using Eqn. (1) to parameterize the Q-function leads to zero approximation error and results in an unbiased estimator. Note that this does not require knowledge of $\phi$. To highlight the significance of Theorem 1, we present the following example, in which the state space cannot be explicitly factored, yet the linear decomposition exists (additional examples probing the sufficient conditions can be found in Appendix C).

**Example 1** (Two-dimensional chains with abstractions). The factored action space shown in Figure 2a, $\mathcal{A} = \mathcal{A}_x \times \mathcal{A}_y$, is the composition of two binary sub-action spaces: $\mathcal{A}_x = \{\leftarrow, \rightarrow\}$ leading the agent to move left or right, and $\mathcal{A}_y = \{\downarrow, \uparrow\}$ leading the agent to move down or up. Thus, $\mathcal{A}$ consists of four actions, where each action $\boldsymbol{a} = [a_x, a_y]$ leads the agent to move ***diagonally***.

Consider the MDP in Figure 2b with action space $\mathcal{A}$. The state space $\mathcal{S} = \{s_{0,0}, s_{0,1}, \tilde{s}_{0,1}, s_{1,0}, s_{1,1}\}$ contains 5 different states; subscripts indicate the abstract state vector under $\phi = [\phi_x, \phi_y]$ (e.g., $s_{0,1}$ and $\tilde{s}_{0,1}$ are two different raw states but are identical under the abstraction, $\phi(s_{0,1}) = \phi(\tilde{s}_{0,1}) = [z_{0,?}, z_{?,1}]$). There does not exist an explicit state space factorization such that $\mathcal{S} = \mathcal{S}_x \times \mathcal{S}_y$. One can check that Eqns. (2) and (3) are satisfied by comparing the raw transitions and rewards against the abstracted version (e.g., action $\nearrow$ from $s_{0,0}$ moves both $\rightarrow$ (under $\phi_x$) and $\uparrow$ (under $\phi_y$) to $s_{1,1}$ and receives the sum of the two rewards, $1 + 1 = 2$). For Eqn. (4) to hold, the policy must take the same action from $s_{0,1}$ and $\tilde{s}_{0,1}$. In Figure 2c, we show the linear decomposition of the Q-function for one such policy where Theorem 1 applies, under which the evolution of the MDP can be seen as two chain MDPs running in parallel (also in Figure 2b). ◁

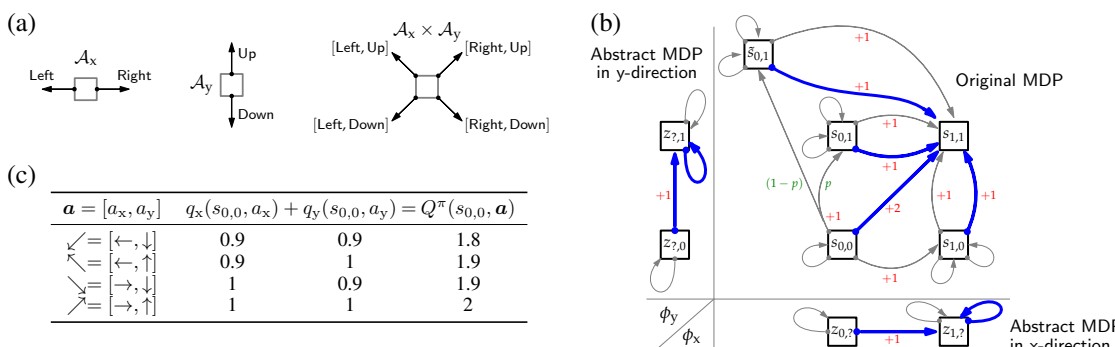

(a) (c) (b)

| $\boldsymbol{a} = [a_\mathrm{x}, a_\mathrm{y}]$ | $q_\mathrm{x}(s_{0,0}, a_\mathrm{x}) + q_\mathrm{y}(s_{0,0}, a_\mathrm{y}) = Q^\pi(s_{0,0}, \boldsymbol{a})$ | | |
|---|---|---|---|
| $\swarrow = [\leftarrow, \downarrow]$ | 0.9 | 0.9 | 1.8 |
| $\nwarrow = [\leftarrow, \uparrow]$ | 0.9 | 1 | 1.9 |
| $\searrow = [\rightarrow, \downarrow]$ | 1 | 0.9 | 1.9 |
| $\nearrow = [\rightarrow, \uparrow]$ | 1 | 1 | 2 |

Figure 2: (a) The composition of sub-action spaces $\mathcal{A}_\mathrm{x}$ and $\mathcal{A}_\mathrm{y}$ results in $\mathcal{A} = \mathcal{A}_\mathrm{x} \times \mathcal{A}_\mathrm{y}$ depicted by outgoing arrows exiting the corners of each state (denoted by $\square$). ***The corner from which the action exits encodes the direction.*** (b) An MDP with 5 states and 4 actions of the factored action space $\mathcal{A}$. For example, action $\nearrow = [\rightarrow, \uparrow]$ from $s_{0,0}$ moves the agent both right ($\rightarrow$) and up ($\uparrow$), to $s_{1,1}$. Under abstractions $\phi = [\phi_\mathrm{x}, \phi_\mathrm{y}]$, this MDP can be mapped to two abstract MDPs (with action spaces $\mathcal{A}_\mathrm{x}$ and $\mathcal{A}_\mathrm{y}$, respectively). The abstract state spaces are $\mathcal{Z}_\mathrm{x} = \{z_{0,?}, z_{1,?}\}$ and $\mathcal{Z}_\mathrm{y} = \{z_{?,0}, z_{?,1}\}$, respectively, where ? indicates the coordinate ignored by the abstraction. $s_{1,1}$ is an absorbing state whose outgoing transition arrows are not shown. Taking action $\nwarrow = [\leftarrow, \uparrow]$ from $s_{0,0}$ leads to $s_{0,1}$ with probability $p$ and to $\tilde{s}_{0,1}$ with probability $(1 - p)$ (denoted in green). Actions taken by a deterministic policy $\pi$ are denoted by **bold blue** arrows. $\pi$ takes the same action $\searrow = [\rightarrow, \downarrow]$ from $s_{0,1}$ and $\tilde{s}_{0,1}$. Nonzero rewards are denoted in red. (c) Linear decomposition of $Q^\pi$ for $s_{0,0}$ with respect to the factored action space ($\gamma = 0.9$). Similar decompositions for other states also exist (omitted for space).

### 3.2 Necessary Conditions for Zero Bias

In Appendix B.5, we derive a necessary condition for the linear parameterization to be unbiased. Unfortunately, the condition is not verifiable unless the exact MDP parameters are known; this highlights the non-trivial nature of the problem. One may naturally question whether the sufficient conditions (which are arguably more verifiable in practice) must hold (i.e., are necessary) for the linear parameterization to be unbiased. Perhaps surprisingly, ***none*** of the conditions are necessary. We state the following propositions and provide justifications through a set of counterexamples below and in Appendix C.

**Proposition 2.** *There exists an MDP $\mathcal{M}$ and a policy $\pi$ for which $Q^\pi_\mathcal{M}$ decomposes as Eqn. (1) but the transition function $p$ of $\mathcal{M}$ does not satisfy Eqn. (2).*

**Proposition 3.** *There exists an MDP $\mathcal{M}$ and a policy $\pi$ for which $Q^\pi_\mathcal{M}$ decomposes as Eqn. (1) but the reward function $r$ of $\mathcal{M}$ does not satisfy Eqn. (3).*

**Proposition 4.** *There exists an MDP $\mathcal{M}$ and a policy $\pi$ for which $Q^\pi_\mathcal{M}$ decomposes as Eqn. (1) but the policy $\pi$ does not satisfy Eqn. (4).*

**Example 2** (Modified two-dimensional chains)**.** In Figure 3, all conditions in Theorem 1 are violated, yet for each state, there exists a linear decomposition of Q-values (see Appendix C). ◁

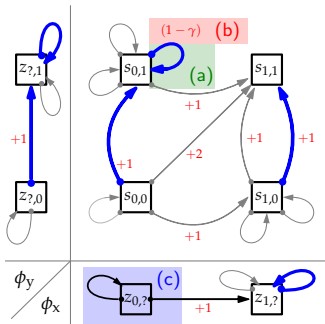

Figure 3: This MDP is similar to Example 1 (except it does not have state $\tilde{s}_{0,1}$) and we consider the same abstractions $\phi = [\phi_\mathrm{x}, \phi_\mathrm{y}]$. The Q-function and decomposition are exactly the same as in the previous example. However, none of the conditions in Theorem 1 are satisfied. (a) The transition function does not satisfy Eqn. (2) because action $\nearrow = [\rightarrow, \uparrow]$ from $s_{0,1}$ does not move right ($\rightarrow$ under $\phi_\mathrm{x}$) to $s_{1,1}$ and instead moves back to state $s_{0,1}$. (b) The reward function does not satisfy Eqn. (3) as the reward of $(1 - \gamma)$ for action $\nearrow = [\rightarrow, \uparrow]$ from $s_{0,1}$ is not the sum of $+1$ ($\rightarrow$ from $z_{0,?}$ under $\phi_\mathrm{x}$) and 0 ($\uparrow$ from $z_{?,1}$ under $\phi_\mathrm{y}$). (c) The policy does not satisfy Eqn. (4) as it takes different sub-actions from $z_{0,?}$ under $\phi_\mathrm{x}$ ($\nwarrow$ from $s_{0,0}$ specifies $\leftarrow$, whereas $\nearrow$ from $s_{0,1}$ specifies $\rightarrow$).

Therefore, while Theorem 1 imposes a rather stringent set of assumptions on the MDP structure (transitions, rewards) and the policy, violations of these conditions do not preclude the linear parameterization of the Q-function from being an unbiased estimator.

### 3.3 How Does Bias Affect Policy Learning?

When the sufficient conditions do not hold perfectly, using the linear parameterization in Eqn. (1) to fit the Q-function may incur nonzero approximation error (bias). This can affect the performance of the learned policy; in Appendix B.3, we derive error bounds based on the extent of bias relative to the sufficient conditions in Theorem 1. Despite this bias, our approach always leads to a reduction in the variance of the estimator. This gives us an opportunity to achieve a better bias-variance trade-off, especially given limited historical data in the offline setting. In addition, as we will demonstrate, biased Q-values do not always result in suboptimal policy performance, and we identify the characteristics of problems where this occurs under our proposed linear decomposition.

#### 3.3.1 Bias-Variance Trade-off

While the amount of bias incurred depends on the problem structure, the benefit of variance reduction is immediate. Intuitively, to learn the Q-function of a tabular MDP with state space $\mathcal{S}$ and action space $\mathcal{A} = \bigotimes_{d=1}^{D} \mathcal{A}_d$, the linear parameterization reduces the number of free parameters from $|\mathcal{S}||\mathcal{A}| = |\mathcal{S}|(\prod_{d=1}^{D} |\mathcal{A}_d|)$ to $|\mathcal{S}|(\sum_{d=1}^{D} |\mathcal{A}_d| - D + 1)$ (see Appendix B.4). This reduces the hypothesis class from exponential in $D$ to linear in $D$. To analyze variance reduction, we compare the bounds on Rademacher complexity [16–18] of the Q-function approximator using the factored action space with that of the full combinatorial action space (formally discussed in Appendix B.6).

**Proposition 5.** *Using the linear Q-function decomposition for the factored action space in Eqn. (1) has a smaller lower bound on the empirical Rademacher complexity compared to learning the Q-function in the combinatorial action space.*

Proposition 5 shows that our linear Q-function parameterization leads to a smaller function space, which implies a lower-variance estimator. Hence, our factored-action approach can make more efficient use of limited samples, leading to an interesting bias-variance trade-off that is especially beneficial for offline settings with limited data.

#### 3.3.2 Bias $\not\Rightarrow$ Suboptimal Performance

Even in the presence of bias, an inaccurate Q-function may still correctly identify the value-maximizing action (Proposition 6). While this statement is generally true, in this section, we identify ***when*** this occurs ***specifically given*** our linear decomposition based on factored action spaces. To focus the analysis on the most interesting aspects unique to our approach, we consider a bandit setting; extensions to the sequential RL setting are possible by applying induction similar to the proof for the main theorems (Appendices B.1 and B.2).

**Proposition 6.** *There exists an MDP with the optimal $Q^*$ and its approximation $\hat{Q}$ parameterized in the form of Eqn. (1), such that $\hat{Q} \neq Q^*$ and yet $\arg\max_{\boldsymbol{a}} \hat{Q}(\boldsymbol{a}) = \arg\max_{\boldsymbol{a}} Q^*(\boldsymbol{a})$.*

*Justification.* Consider a 1-step bandit problem with a single state and the same action space as before, $\mathcal{A} = \mathcal{A}_{\text{x}} \times \mathcal{A}_{\text{y}}$. Taking an action $\boldsymbol{a} = [a_{\text{x}}, a_{\text{y}}]$ leads the agent to move diagonally and terminate immediately. Since there are no transitions, the Q-values of any policy are simply the immediate reward from each action, $Q(\boldsymbol{a}) = r(\boldsymbol{a})$. We assume the reward function is defined as in Figure 4a (Appendix B.7 describes a procedure to standardize an arbitrary reward function).

(a)

(b)

$$r_{\text{Left}} + r_{\text{Down}} = 0$$
$$r_{\text{Left}} + r_{\text{Up}} = \alpha$$
$$r_{\text{Right}} + r_{\text{Down}} = 1$$
$$r_{\text{Right}} + r_{\text{Up}} + r_{\text{Interact}} = 1 + \alpha + \beta$$

(c)

True Value Function    Linear Approximation

$$\begin{bmatrix} Q^*(\swarrow) \\ Q^*(\nwarrow) \\ Q^*(\searrow) \\ Q^*(\nearrow) \end{bmatrix} = \begin{bmatrix} 0 \\ \alpha \\ 1 \\ 1 + \alpha + \beta \end{bmatrix} \qquad \begin{bmatrix} \hat{Q}(\swarrow) \\ \hat{Q}(\nwarrow) \\ \hat{Q}(\searrow) \\ \hat{Q}(\nearrow) \end{bmatrix} = \begin{bmatrix} -\frac{1}{4}\beta \\ \alpha + \frac{1}{4}\beta \\ 1 + \frac{1}{4}\beta \\ 1 + \alpha + \frac{3}{4}\beta \end{bmatrix}$$

Figure 4: (a) A two-dimensional bandit problem with action space $\mathcal{A}$. Rewards are denoted for each arm. (b) Learning using the linear Q decomposition approach corresponds to a system of linear equations that relates the reward of each arm. The parameter $r_{\text{Interact}}$ is dropped in our linear approximation, leading to omitted-variable bias. (c) Solving the system results in an approximate value function $\hat{Q}$, which does not equal to the true value function $Q^*$ unless $\beta = 0$.

Applying our approach amounts to solving for the parameters $r_{\text{Left}}, r_{\text{Right}}, r_{\text{Down}}, r_{\text{Up}}$ of the linear system in Figure 4b, while dropping the interaction term $r_{\text{Interact}}$, resulting in a form of omitted-variable bias [19]. Solving the system gives the approximate value function where the interaction term $\beta$ appears in the approximation $\hat{Q}$ for all arms (Figure 4c, details in Appendix B.8).

Note that $\hat{Q} = Q^*$ only when $\beta = 0$, i.e., there is no interaction between the two sub-actions. We first consider the family of problems with $\alpha = 1$ and $\beta \in [-4, 4]$. In Figure 5a, we measure the value approximation error $\text{RMSE}(Q^*, \hat{Q})$, as well as the suboptimality $V^{\pi^*} - V^{\hat{\pi}} = \max_{\boldsymbol{a}} Q^*(\boldsymbol{a}) - Q^*(\arg\max_{\boldsymbol{a}} \hat{Q}(\boldsymbol{a}))$ of the greedy policy defined by $\hat{Q}$ as compared to $\pi^*$. As expected, when $\beta = 0$, $\hat{Q}$ is unbiased and has zero approximation error. When $\beta \neq 0$, $\hat{Q}$ is biased and $\text{RMSE} > 0$; however, for $\beta \geq -1$, $\hat{Q}$ corresponds to a policy that correctly identifies the optimal action.

We further investigate this phenomenon considering both $\alpha, \beta \in [-4, 4]$ (to show all regions with interesting trends), measuring RMSE and suboptimality in the same way as above. As shown in Figure 5b, the approximation error is zero only when $\beta = 0$, regardless of $\alpha$. However, in Figure 5c, for a wide range of $\alpha$ and $\beta$ settings, suboptimality is zero; this suggests that in those regions, even in the presence of bias (non-zero approximation error), our approach leads to an approximate value function that correctly identifies the optimal action. The irregular contour outlines multiple regions where this happens; one key region is when the two sub-actions affect the reward in the same direction (i.e., $\alpha \geq 0$) and their interaction effects also affect the reward in the same direction (i.e., $\beta \geq 0$).

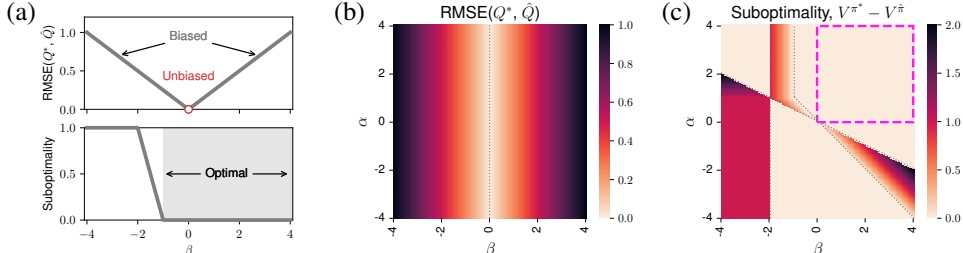

Figure 5: (a) The approximation error and policy suboptimality of our approach for the bandit problem in Figure 4a, for different settings of $\beta$ when $\alpha = 1$. The Q-value approximation is unbiased only when $\beta = 0$, but the corresponding approximate policy is optimal for a wider range of $\beta \geq -1$. (b-c) The approximation error and policy suboptimality of our approach for the bandit problem in Figure 4a, for different settings of $\alpha$ and $\beta$. The Q-value approximation is unbiased only when $\beta = 0$, but the corresponding approximate policy is optimal for a wide range of $\alpha$ and $\beta$ values. The highlighted region of zero suboptimality corresponds to $\alpha \geq 0$ and $\beta \geq 0$.

### 3.4 Practical Considerations: Are these Assumptions too Strong?

Based on our theoretical analysis, strong assumptions (Section 3.1) on the problem structure (though not necessary, Section 3.2) are the only known way to guarantee the unbiasedness of our proposed linear approximation. It is thus crucial to understand the applicability (and inapplicability) of our approach in real-world scenarios. Exploring to what extent these assumptions hold in practice is especially important for safety-critical domains such as healthcare where incorrect actions (treatments) can have devastating consequences. Fortunately, RL tasks for healthcare are often equipped with significant domain knowledge, which serves as a better guide to inform the algorithm design than heuristics-driven reasoning alone [20, 5, 9].

Oftentimes, when clinicians treat conditions using multiple medications at the same time (giving rise to the factored action space), it is because each medication has a different "mechanism of action," resulting in negligible or limited interactions. For example, several classes of medications are used in the management of chronic heart failure, and each has a unique and incremental benefits on patient outcomes [21]. Problems such as this satisfy the sufficient conditions in Section 3.1 in spite of a non-factorized state space. Moreover, any small interactions would have a bounded effect on RL policy performance (according to Appendix B.3).

Similarly, in the management of sepsis (which we consider in Section 4.2), fluids and vasopressors affect blood pressure to correct hypotension via different mechanisms [22]. Fluid infusion increases "preload" by increasing the blood return to the heart to make sure the heart has enough blood to pump out [23]. In contrast, common vasopressors (e.g., norepinephrine) increase "inotropy" by

stimulating the heart muscle and increase peripheral vascular resistance to maintain perfusion to organs [24, 25]. Therefore, while the two treatments may appear to operate on the same part of the state space (e.g., they both increase blood pressure), in general they are not expected to interfere with each other. Recently, there has also been evidence suggesting that their combination can better correct hypotension [26], which places this problem approximately in the regime discussed in Section 3.3.2.

In offline settings with limited historical data, the benefits of a reduction in variance can outweigh any potential small bias incurred in the scenarios above and lead to overall performance improvement (Section 3.3.1). However, our approach is not suitable if the interaction is counter to the effect of the sub-actions (e.g., two drugs that raise blood pressure individually, but when combined lead to a decrease). In such scenarios, the resulting bias will likely lead to suboptimal performance (Section 3.3.2). Nevertheless, many drug-drug interactions are known and predictable [27–30]. In such cases, one can either explicitly encode the interaction terms or resort back to a combinatorial action space (Appendix B.9). While we focus on healthcare, there are other domains in which significant domain knowledge regarding the interactions among sub-actions is available, e.g., cooperative multi-agent games in finance where there is a higher payoff if agents cooperate (positive interaction effects) or intelligent tutoring systems that teach basic arithmetic operations as well as fractions (which are distinct but related skills). For these problems, this knowledge can and should be leveraged.

## 4 Experimental Evaluations

We apply our approach to two offline RL problems from healthcare: a simulated and a real-data problem, both having an action space that is composed of several sub-action spaces. These problems correspond to settings discussed in Section 3.4 where we expect our proposed approach to perform well. In the following experiments, we compare our proposed approach (Figure 1b), which makes assumptions regarding the effect of sub-actions in combination with other sub-actions, against a common baseline that considers a combinatorial action space (Figure 1a).

### 4.1 Simulated Domain: Sepsis Simulator

**Rationale.** First, we apply our approach to a simulated domain modeled after the physiology of patients with sepsis [13]. Although the policies are learned "offline," a simulated setting allows us to evaluate the learned policies in an "online" fashion without requiring offline policy evaluation (OPE).

**Setup.** Following prior work [31], a state is represented by a feature vector $\mathbf{x}(s) \in \{0, 1\}^{21}$ that uses a one-hot encoding for each underlying variable (diabetes status, heart rate, blood pressure, oxygen concentration, glucose; all of which are discrete). The action space is composed of 3 binary treatments: antibiotics, vasopressors, and mechanical ventilation, such that $\mathcal{A} = \mathcal{A}_{\text{abx}} \times \mathcal{A}_{\text{vaso}} \times \mathcal{A}_{\text{mv}}$, with $\mathcal{A}_{\text{abx}} = \mathcal{A}_{\text{vaso}} = \mathcal{A}_{\text{mv}} = \{0, 1\}$ and $|\mathcal{A}| = 2^3 = 8$. Each treatment affects certain vital signs and may raise or lower their values with pre-specified probabilities (precise definition in [31]). A patient is discharged alive when all vitals are normal and all treatments have been withdrawn; death occurs if 3 or more vitals are abnormal. Rewards are sparse and only assigned at the end of each episode ($+1$ for survival and $-1$ for death), after which the system transitions into the respective absorbing state. Episodes are truncated at a maximum length of 20 following [13] (where no terminal reward is assigned). Here, the MDP partly satisfies the sufficient conditions outlined in Section 3. For example, oxygen saturation (which can be seen as a state abstraction) is only affected by mechanical ventilation, whereas heart rate is only affected by antibiotics. However, blood pressure is affected by both antibiotics and vasopressors, meaning the effects of these two sub-actions are *not* independent.

**Offline learning.** First, we generated datasets with different sample sizes following different behavior policies. We ran fitted Q-iteration for up to 50 iterations using a neural network function approximator, selecting the early-stopping iteration based on ground-truth policy performance. Each setting of sample size and behavior policy was repeated 10 times with different random seeds. Additional details are described in Appendix D.1.

**Results.** Figure 6 compares median performance of the proposed approach vs. the baseline over the 10 runs (error bars are interquartile ranges). We considered behavior policies that take the optimal action with probability $\rho$ and select randomly among non-optimal actions with probability $1 - \rho$.

*How does sample size affect performance?* We first look at a uniformly random behavior policy ($\rho = 1/|\mathcal{A}| = 0.125$, Figure 6 center). As expected, larger sample sizes (i.e., more training episodes) lead to better policy performance for both the baseline and proposed approaches. For smaller

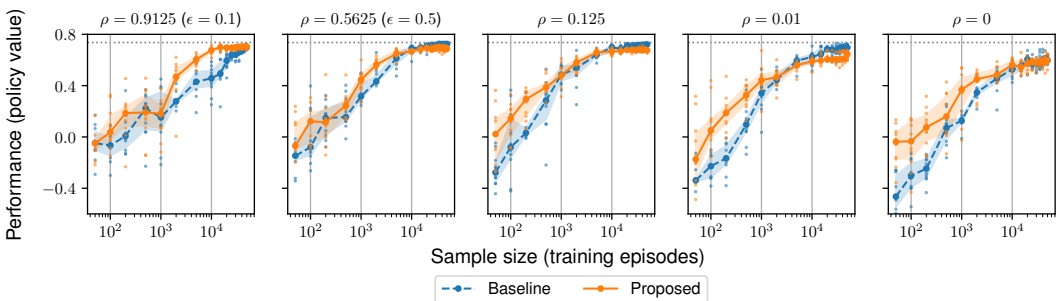

Figure 6: Performance on the sepsis simulator across sample sizes and behavior policies. Plots display the performance over 10 runs, with the trend lines showing medians and error bars showing interquartile ranges. $\rho$ is the probability of taking the optimal action under the behavior policies used to generate offline datasets from the simulator. The left two plots show two $\epsilon$-greedy policies ($\rho > 0.125$; conversion: $\rho = (1 - \epsilon) + \epsilon/|\mathcal{A}|$); the middle plot shows a uniformly random policy ($\rho = 0.125$); the right two plots show two policies that undersample the optimal action, $\rho < 0.125$; from left to right, $\rho$ decreases. Across different data distributions, our proposed approach outperforms the baseline at small sample sizes, and closely matches baseline performance at large sample sizes. Dashed lines denote the value of the optimal policy, which equals to 0.736.

sample sizes ($< 5000$), the proposed approach consistently outperforms the baseline. As sample size increases further, the performance gap shrinks and eventually the baseline overtakes our proposed approach. This is because variance decreases with increasing sample size but the bias incurred by the factored approximation does not change. Once there are enough samples, reductions in variance are no longer advantageous and the incurred bias dominates the performance. Overall, this shows that our approach is promising especially for datasets with limited sample size.

*How does behavior policy affect performance?* As we anneal the behavior policy closer to the optimal policy ($\rho > 0.125$, Figure 6 left two), we reduce the randomness in the behavior policy and limit the amount of exploration possible at the same sample size. The same overall trend largely holds. On the other hand, when the probability of taking the optimal action is less than random ($\rho < 0.125$, Figure 6 right two), the proposed approach achieves better performance than the baseline with an even larger gap for limited sample sizes ($\leq 10^3$). Without observing the optimal actions ($\rho = 0$), the baseline performs relatively poorly, even for large sample sizes. In comparison, our approach accounts for relationships among actions to some extent and is thus able to better generalize to the unobserved and underexplored optimal actions, thereby outperforming the baseline.

**Takeaways.** In a challenging situation where our theoretical assumptions do not perfectly hold, our proposed approach matches or outperforms the baseline, especially for smaller sample sizes.

### 4.2 Real Healthcare Data: Sepsis Treatment in MIMIC-III

**Rationale.** We apply our method to a real-world example of learning optimal sepsis treatment policies for patients in the intensive care unit. Acknowledging the challenging nature of OPE for quantitative comparisons [32, 33], here we qualitatively inspect the learned policies using clinical domain knowledge.

**Setup.** Originally introduced by [2], we use the improved formulations of this task as per [34] and [35]. After applying the specified inclusion and exclusion criteria to the de-identified MIMIC-III database [36], we obtained a cohort of 19,287 patients and performed a 70/15/15 split for training, validation and testing. For each patient, their data include 10 time-invariant demographic and contextual features and a 33-dimensional time series collected at 4h intervals, consisting of measurements from up to 24h before until up to 48h after sepsis onset. We used a recurrent neural network (RNN) with long short-term memory (LSTM) cells to create an approximate information state [37] to summarize the history into a $d_{\mathcal{S}}$-dimensional embedding vector. A terminal reward of 100 is assigned for 48h survival and 0 otherwise. Intermediate rewards are all 0. $\gamma$ for learning is 0.99 and for evaluation is 1. Actions pertain to treatment decisions in each 4h interval, representing total volume of intravenous (IV) fluids and amount of vasopressors administered, resulting in a $5 \times 5$ factored action space.

**Offline learning.** After learning the state representations, we apply variants of discrete-action batch-constrained Q-learning (BCQ) [38, 14], where the baseline uses the combinatorial action space and the proposed approach incorporates the linear decomposition induced by the factored action space.

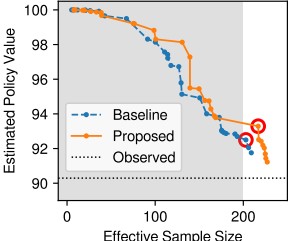

| Policy | Baseline BCQ | Factored BCQ | Clinician |
|---|---|---|---|
| Test WIS | $90.44 \pm 2.44$ | $91.62 \pm 2.12$ | $90.29 \pm 0.51$ |
| Test ESS | $178.32 \pm 11.42$ | $178.32 \pm 11.96$ | 2894 |
| % agreement with clinician | 62.42% | 62.37% | 57.16% |

Figure 7: Left - Pareto frontiers of validation performance for the candidate policies (all points plotted in Figure 16). The shaded region does not meet the ESS cutoff of $\geq 200$. The red circles indicate the selected models (based on best validation WIS) for baseline and proposed (both have a BCQ threshold of $\tau = 0.5$). Right - Performance on test set, $\pm$ standard errors from 100 bootstraps.

(a)

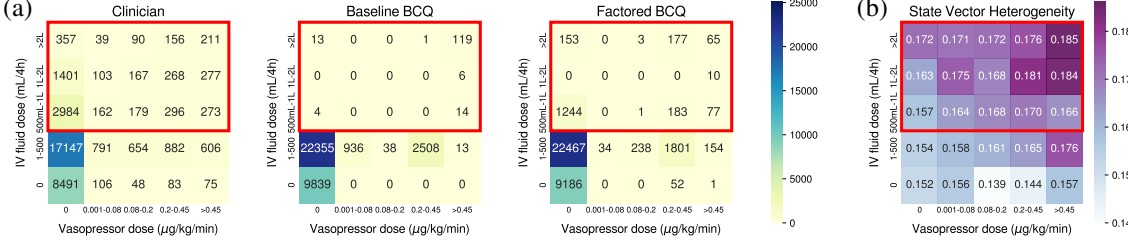

(b)

Figure 8: (a) Qualitative comparison of policies. (b) Per-action state heterogeneity, measured as the standard deviation of all state embeddings from which a particular action is observed in the dataset, averaged over state embedding dimensions. Actions with higher IV fluid doses exhibited greater heterogeneity in the observed states from which those actions were taken (by the clinician policy).

The Q-networks were trained for a maximum of $10,000$ iterations, with checkpoints saved every 100 iterations. We performed model selection [31] over the saved checkpoints (candidate policies) by evaluating policy performance using the validation set with OPE. Specifically, we estimated policy value using weighted importance sampling (WIS) and measured effective sample size (ESS), where the behavior policy was estimated using $k$ nearest neighbors in the embedding space. Following previous work [39], the final policies were selected by maximizing validation WIS with ESS of $\geq 200$ (we consider other thresholds in Appendix D.3), for which we report results on the test set.

**Results.** We visualize the validation performance over all candidate policies. Figure 7-left shows that the performance Pareto frontier (in terms of WIS and ESS) of the proposed approach generally dominates the baseline.

*Quantitative comparisons.* Evaluating the final selected policies on the test set (Figure 7-right) shows that the proposed factored BCQ achieves a higher policy value (estimated using WIS) than baseline BCQ at the same level of ESS. In addition, both policies have a similar level of agreement with the clinician policy, comparable to the average agreement among clinicians.

*Qualitative comparisons.* In Figure 8a, we compare the distributions of recommended actions by the clinician behavior policy, baseline BCQ and factored BCQ, as evaluated on the test set. While overall the policies look rather similar, in that the most frequently recommended action corresponds to low doses of IV fluids <500mL with no vasopressors, there are notable differences for key parts of the action space. In particular, baseline BCQ almost never recommends higher doses of IV fluids >500 mL, either alone or in combination with vasopressors, whereas both clinician and factored BCQ recommend IV fluids >500 mL more frequently. These actions are typically used for critically ill patients, for whom the Surviving Sepsis Campaign guidelines recommends up to >2L of fluids [40]. We hypothesize that this difference is due to a higher level of heterogeneity in the patient states for which actions with high IV fluid doses were observed, compared to the remaining actions with lower doses of IV fluids. To further understand this phenomenon, we measure the per-action state heterogeneity in the test set by computing, for each action, the standard deviation (averaged over the embedding dimensions) of all RNN state embeddings from which that action is taken according to the behavior policy. As shown in Figure 8b, actions with higher IV fluids generally have larger standard deviations, supporting our hypothesis. The larger heterogeneity combined with lower sample sizes makes it difficult for baseline BCQ to correctly infer the effects of these actions, as it does not leverage the relationship among actions. In contrast, our approach leverages the factored action space and can thus make better inferences about these actions.

**Takeaways.** Applied to real clinical data, our proposed approach outperforms the baseline quantitatively and recommends treatments that align better with clinical knowledge. While promising, these results are based in part on OPE, which has many issues [32, 33]. We stress that further investigation and close collaboration with clinicians are essential before such RL algorithms are used in practice.

## 5 Related Work

For many years, the factored RL literature focused exclusively on state space factorization [15, 41–43]. More recently, interest in action space factorization has grown, as RL is applied in increasingly more complex planning and control tasks. In particular, researchers have previously considered the model-based setting with known MDP factorizations in which both state and action spaces are factorized [44–47]. For model-free approaches, others have studied methods for factored actions with a policy component (i.e., policy-based or actor-critic) [48, 20, 49–51]. In contrast, our work considers value-based methods as those have been the most successful in offline RL [52].

Among prior work with a value-based component (e.g., Q-network), the majority pertains to multi-agent [8–10, 53, 54] or multi-objective [51] problems that impose known, explicit assumptions on the state space or the reward function. Notably, Son et al. [10] established theoretical conditions for factored optimal actions (called "Individual-Global-Max") for multi-agent RL and motivated subsequent works [55, 56]; their result differs from our contribution on the unbiasedness of factored Q-functions (instead of actions) for single-agent RL. In the online setting for single-agent deep RL, Sharma et al. [20] and Tavakoli et al. [5] incorporated factored action spaces into Q-network architecture designs, but did not provide a formal justification for the linear decomposition. Others have empirically compared various "mixing" functions to combine the values of sub-actions [20, 9]. In contrast, while our work only considers the linear decomposition function, we examine its theoretical properties and provide justifications for using this approach in practical problems, especially in offline settings. Our linear Q-decomposition is related to that of Swaminathan et al. [57] who also applied a linearity assumption for off-policy evaluation, but for combinatorial contextual bandits rather than RL. Our insights on the bias-variance trade-off is also related to a concurrent work by Saito and Joachims [58] who proposed efficient off-policy evaluation for bandit problems with large (but not necessarily factored) action spaces. In appendix Table 1, we further outline the differences of our work compared to the existing literature.

Finally, the sufficient conditions we establish are related to, but different from, those identified by Van Seijen et al. [59] and Juozapaitis et al. [12] who considered reward decompositions in the absence of factored actions. Related, Metz et al. [60] proposed an approach that sequentially predicts values for discrete dimensions of a transformed continuous action space, but assume an *a priori* ordering of action dimensions, which we do not; Pierrot et al. [50] studied a different form of action space factorization where sub-actions are sequentially selected in an autoregressive manner. Complementary to our work, Tavakoli et al. [6] proposed to organize the sub-actions and interactions as a hypergraph and linearly combining the values; our theoretical results on the linear decomposition nonetheless apply to their setting where the sub-action interactions are explicitly identified and encoded.

## 6 Conclusion

To better leverage factored action spaces in RL, we developed an approach to learning policies that incorporates a simple linear decomposition of the Q-function. We theoretically analyze the sufficient and necessary conditions for this parameterization to yield unbiased estimates, study its effect on variance reduction, and identify scenarios when any resulting bias does not lead to suboptimal performance. We also note how domain knowledge may be used to inform the applicability of our approach in practice, for problems where any possible bias is negligible or does not affect optimality. Through empirical experiments on two offline RL problems involving a simulator and real clinical data, we demonstrate the advantage of our approach especially in settings with limited sample sizes. We provide further discussions on limitations, ethical considerations and societal impacts in Appendix A. Though motivated by healthcare, our approach could apply more broadly to scale RL to other applications (e.g., education) involving combinatorial action spaces where domain knowledge may be used to verify the theoretical conditions. Future work should consider the theoretical implications of linear Q decompositions when combined with other offline RL-specific algorithms [52]. Given the challenging nature of identifying the best treatments from offline data, our proposed approach may also be combined with other RL techniques that do not aim to identify the single best action (e.g., learning dead-ends [61] or set-valued policies [34]).

## Acknowledgments

This work was supported by the National Science Foundation (NSF; award IIS-1553146 to JW; award IIS-2007076 to FDV; award IIS-2153083 to MM) and the National Library of Medicine of the National Institutes of Health (NLM; grant R01LM013325 to JW and MWS). The views and conclusions in this document are those of the authors and should not be interpreted as necessarily representing the official policies, either expressed or implied, of the National Science Foundation, nor of the National Institutes of Health. This work was supported, in part, by computational resources and services provided by Advanced Research Computing, a division of Information and Technology Services (ITS) at the University of Michigan, Ann Arbor. The authors would like to thank Adith Swaminathan, Tabish Rashid, and members of the MLD3 group for helpful discussions regarding this work, as well as the reviewers for constructive feedback.

## Data and Code Availability

The code for all experiments is available at https://github.com/MLD3/OfflineRL_FactoredActions. The sepsis simulator is based on prior work with public implementation at https://github.com/clinicalml/gumbel-max-scm. The MIMIC-III database used in the real-data experiments of this paper is publicly available through the PhysioNet website: https://physionet.org/content/mimiciii/1.4/. The cohort definition, extraction and preprocessing code are based on prior work with publicly available implementation at https://github.com/microsoft/mimic_sepsis.

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
