# A  Additional Discussion

**Computational Efficiency.** While our main analysis focuses on statistical efficiency (variance) and its trade-off with approximation error (bias), here we outline some considerations on computational efficiency. To compute the values for all output heads in Figure 1, there is a clear saving of computational cost by our approach with a linear complexity $O(D)$ (measured in flops) in the number of sub-actions, whereas the baseline has an exponential complexity $O(\exp(D))$. We consider two common inference operations after the values of the output heads are computed: $\max_{\boldsymbol{a}} Q(s, \boldsymbol{a})$ and $\arg\max_{\boldsymbol{a}} Q(s, \boldsymbol{a})$. For both operations, the baseline has an exponential time complexity of $O(\exp(D))$. For our proposed approach, an optimized implementation has a linear time complexity of $O(D)$: one can perform argmax/max per sub-action and then concatenate/sum the results. In our current code release, we did not implement the optimized version; instead, we made use of the sub-action featurization matrix defined in Appendix B.4 so that automatic differentiation can be applied directly. This implementation is computationally more expensive than our analysis above and than the baseline: the forward pass includes a dense matrix multiplication with time complexity $O(D \exp(D))$ flops, followed by an $O(\exp(D))$ argmax/max operation. In settings where computational complexity might be a bottleneck (especially at inference time), we recommend using the featurization matrix implementation for learning and the optimized version for inference.

**Limitations.** Our theoretical analysis in Section 3 focuses on the "realizability" condition of the linear function class [62], where we are interested in guarantees of zero approximation error, i.e., whether the true $Q^*$ lies within the linear function class. In principle, it is possible to find $Q^*$ given a realizable function class (e.g., by enumerating all member functions). However, when Q-learning-style iterative algorithms are used in practice, its convergence relies on a stronger "completeness" condition, as discussed in [62–64]. We did not investigate how our proposed form of parameterization (and the specific shape of bias introduced) interacts with the learning procedure, and this is an interesting direction for future work (Wang et al. [56] studied this for linear value factorization in the context of FQI but for multi-agent RL).

**Ethical Considerations and Societal Impact.** In general, policies computationally derived using RL must be carefully validated before they can be used in high-stakes domains such as healthcare. Our linear parameterization implicitly makes an independence assumption with respect to the sub-actions, allowing the Q-function to generalize to sub-action combinations that are underexplored (and even unexplored) in the offline data (as shown in Section 4.1). When the independence assumptions are valid (according to domain knowledge), this is a case of a "free lunch" as we can reduce variance without introducing any bias. However, inaccurate or incomplete domain knowledge may render the independence assumptions invalid and cause the agent to incorrectly generalize to dangerous actions (e.g., learned policy recommends drug combinations with adverse side effects, see Section 3.4). This misuse may be alleviated by incorporating additional offline RL safeguards to constrain the learned policy (e.g., BCQ was used in Section 4.2 to restrict the learned policy to not take rarely observed sub-action combinations). Still, to apply RL in healthcare and other safety-critical domains, it is important to consult and closely collaborate with domain experts (e.g., clinicians for healthcare problems) to come up with meaningful tasks and informed assumptions, and perform thorough evaluations involving both the quantitative and qualitative aspects [32, 33].

Table 1: Qualtitative comparisons of this work with the existing literature.

| | Policy-based? | Model-based? | Value-based? | Linear value decomposition? | Known state factorization or abstraction? | Unbiasedness guarantees? |
|---|---|---|---|---|---|---|
| [46, 47] | | ✓ | | | ✓ | |
| [48] | ✓ | | | | ✓ | |
| [50, 51] | ✓ | | | | ✗ | |
| [15, 41–43] | | | ✓(V) | ✓ | ✓ | |
| [20] | ✓ | | ✓(Q) | ✓¹ | ✓ | |
| VDN [8] | | | ✓(Q) | ✓ | ✓ | |
| QMIX [9] | | | ✓(Q) | *² | ✓ | |
| BDQ [5] | | | ✓(Q) | *² | ✗ | |
| **This work** | | | ✓(Q) | ✓ | ✗ | ✓ |

[1]Empirically tested various "combination" functions including linear. [2]Both QMIX and BDQ do not aggregate the sub-Q functions; instead, they aggregate the argmax sub-actions.

# B Detailed Theoretical Analyses

## B.1 Sufficient Condition: The Trivial Setting - $D$ Parallel MDPs

To build intuition, we first consider a related setting where $D$ MDPs are running in parallel. If every MDP evolves independently as controlled by its respective policy, then the total return from all $D$ MDPs should naturally be the sum of the individual returns from each MDP. Formally, we state the following proposition involving fully factored MDPs and factored policies. Here, we use the vector notation $\boldsymbol{s} = [s_1, \cdots, s_D]$ to indicate the explicit state space factorization.

**Definition 1.** Given MDPs $\mathcal{M}_1 \cdots \mathcal{M}_D$ where each $\mathcal{M}_d$ is defined by $(\mathcal{S}_d, \mathcal{A}_d, p_d, r_d)$, a fully factored MDP $\mathcal{M} = \bigotimes_{d=1}^D \mathcal{M}_d$ is defined by $\mathcal{S}, \mathcal{A}, p, r$ such that $\mathcal{S} = \bigotimes_{d=1}^D \mathcal{S}_d$, $\mathcal{A} = \bigotimes_{d=1}^D \mathcal{A}_d$, $p(\boldsymbol{s}'|\boldsymbol{s}, \boldsymbol{a}) = \prod_{d=1}^D p_d(s_d'|s_d, a_d)$, and $r(\boldsymbol{s}, \boldsymbol{a}) = \sum_{d=1}^D r_d(s_d, a_d)$.

**Definition 2.** Given MDPs $\mathcal{M}_1 \cdots \mathcal{M}_d$ and policies $\pi_1 \cdots \pi_D$ where each $\pi_d : \mathcal{S}_d \rightarrow \Delta(\mathcal{A}_d)$, then a factored policy $\pi = \bigotimes_{d=1}^D \pi_j$ for the MDP $\mathcal{M} = \bigotimes_{d=1}^D \mathcal{M}_j$ is $\pi : \mathcal{S} \rightarrow \Delta(\mathcal{A})$ such that $\pi(\boldsymbol{a}|\boldsymbol{s}) = \prod_{d=1}^D \pi_d(a_d|s_d)$.

**Proposition 7.** *The Q-function of policy $\pi = \bigotimes_{d=1}^D \pi_j$ for MDP $\mathcal{M} = \bigotimes_{d=1}^D \mathcal{M}_d$ can be expressed as $Q_{\mathcal{M}}^{\pi}(\boldsymbol{s}, \boldsymbol{a}) = \sum_{d=1}^D Q_{\mathcal{M}_d}^{\pi_d}(s_d, a_d)$.*

To match the form in Eqn. (1), we can set $q_d(\boldsymbol{s}, a_d) = Q_{\mathcal{M}_d}^{\pi_d}(s_d, a_d)$. Importantly, each $Q_{\mathcal{M}_d}^{\pi_d}$ does not depend on any $a_{d'}$ where $d' \neq d$. Note that although our definition of $q_d$ is allowed to condition on the entire state space $\boldsymbol{s}$, each $Q_{\mathcal{M}_d}^{\pi_d}$ only depends on $s_d$. Proposition 7 can be seen as a corollary to Theorem 1 where the abstractions are defined using the sub-state spaces, such that $\phi_d : \mathcal{S} \rightarrow \mathcal{S}_d$.

*Proof of Proposition 7.* Without loss of generality, we consider the setting with $D = 2$ such that $\mathcal{A} = \mathcal{A}_1 \times \mathcal{A}_2$; extension to $D > 2$ is straightforward. The proof is based on mathematical induction on a sequence of $h$-step Q-functions of $\pi$ defined as $Q_{\mathcal{M}}^{\pi,(h)}(\boldsymbol{s}, \boldsymbol{a}) = \mathbb{E}[\sum_{t=1}^h \gamma^{t-1} r_t | \boldsymbol{s}_1 = \boldsymbol{s}, \boldsymbol{a}_1 = \boldsymbol{a}, \boldsymbol{a}_t \sim \pi]$.

*Base case.* For $h = 1$, the one-step Q-function is simply the reward, which by assumption $r(\boldsymbol{s}, \boldsymbol{a}) = r_1(s_1, a_1) + r_2(s_2, a_2)$. Therefore, $Q_{\mathcal{M}}^{\pi,(1)}(s, \boldsymbol{a}) = Q_{\mathcal{M}_1}^{\pi_1,(1)}(s_1, a_1) + Q_{\mathcal{M}_2}^{\pi_2,(1)}(s_2, a_2)$.

*Inductive step.* Suppose $Q_{\mathcal{M}}^{\pi,(h)}(\boldsymbol{s}, \boldsymbol{a}) = Q_{\mathcal{M}_1}^{\pi_1,(h)}(s_1, a_1) + Q_{\mathcal{M}2}^{\pi_2,(h)}(s_2, a_2)$ holds. We can express $Q_{\mathcal{M}}^{\pi,(h+1)}$ in terms of $Q_{\mathcal{M}}^{\pi,(h)}$ using the Bellman equation:

$$Q_{\mathcal{M}}^{\pi,(h+1)}(\boldsymbol{s}, \boldsymbol{a}) = \underbrace{r(\boldsymbol{s}, \boldsymbol{a})}_{①} + \gamma \underbrace{\sum_{\boldsymbol{s}'} p(\boldsymbol{s}'|\boldsymbol{s}, \boldsymbol{a}) V_{\mathcal{M}}^{\pi,(h)}(\boldsymbol{s}')}_{②}$$

where $V_{\mathcal{M}}^{\pi,(h)}(\boldsymbol{s}') = \sum_{\boldsymbol{a}'} \pi(\boldsymbol{a}'|\boldsymbol{s}') Q_{\mathcal{M}}^{\pi,(h)}(\boldsymbol{s}', \boldsymbol{a}')$.

By Definition 1, ① can be written as a sum $r(\boldsymbol{s}, \boldsymbol{a}) = r_1(s_1, a_1) + r_2(s_2, a_2)$ where each summand depends on only either $a_1$ or $a_2$ but not both. Next we show that ② also decomposes in a similar manner. For a given $\boldsymbol{s}$ we have:

$$V_{\mathcal{M}}^{\pi,(h)}(\boldsymbol{s}) = \sum_{\boldsymbol{a}} \pi(\boldsymbol{a}|\boldsymbol{s}) Q_{\mathcal{M}}^{\pi,(h)}(\boldsymbol{s}, \boldsymbol{a})$$

$$= \sum_{a_1, a_2} \pi_1(a_1|s_1) \pi_2(a_2|s_2) \left( Q_{\mathcal{M}_1}^{\pi_1,(h)}(s_1, a_1) + Q_{\mathcal{M}_2}^{\pi_2,(h)}(s_2, a_2) \right)$$

$$= \left( \cancel{\sum_{a_2} \pi_2(a_2|s_2)}^{1} \right) \sum_{a_1} \pi_1(a_1|s_1) Q_{\mathcal{M}_1}^{\pi_1,(h)}(s_1, a_1) + \left( \cancel{\sum_{a_1} \pi_1(a_1|s_1)}^{1} \right) \sum_{a_2} \pi_2(a_2|s_2) Q_{\mathcal{M}_2}^{\pi_2,(h)}(s_2, a_2)$$

$$= \left( \sum_{a_1} \pi_1(a_1|s_1) Q_{\mathcal{M}_1}^{\pi_1,(h)}(s_1, a_1) \right) + \left( \sum_{a_2} \pi_2(a_2|s_2) Q_{\mathcal{M}_2}^{\pi_2,(h)}(s_2, a_2) \right),$$

where we use the fact that $\pi_1(a_1|s_1)Q_{\mathcal{M}_1}^{\pi_1,(h)}(s_1,a_1)$ is independent of $\pi_2(a_2|s_2)$ (and vice versa), and that $\pi_d(\cdot|s_d)$ is a probability simplex. Letting $V_{\mathcal{M}_d}^{\pi_d,(h)}(s_d) = \sum_{a_d} \pi_1(a_d|s_d)Q_{\mathcal{M}_d}^{\pi_d,h}(s_d,a_d)$, then $V_{\mathcal{M}}^{\pi,(h)}(s') = V_{\mathcal{M}_1}^{\pi_1,(h)}(s_1') + V_{\mathcal{M}_2}^{\pi_2,(h)}(s_2')$.

Substituting into ②, we have:

$$\sum_{s'} p(s'|s,a)V_{\mathcal{M}}^{\pi,(h)}(s')$$

$$= \sum_{s_1',s_2'} p_1(s_1'|s_1,a_1)p_2(s_2'|s_2,a_2)\left(V_{\mathcal{M}_1}^{\pi_1,(h)}(s_1') + V_{\mathcal{M}_2}^{\pi_2,(h)}(s_2')\right)$$

$$= \left(\overbrace{\sum_{s_2'} p_2(s_2'|s_2,a_2)}^{1}\right)\sum_{s_1'} p_1(s_1'|s_1,a_1)V_{\mathcal{M}_1}^{\pi_1,(h)}(s_1') + \left(\overbrace{\sum_{s_1'} p_1(s_1'|s_1,a_1)}^{1}\right)\sum_{s_2'} p_2(s_2'|s_2,a_2)V_{\mathcal{M}_2}^{\pi_2,(h)}(s_2')$$

$$= \left(\sum_{s_1'} p_1(s_1'|s_1,a_1)V_{\mathcal{M}_1}^{\pi_1,(h)}(s_1')\right) + \left(\sum_{s_2'} p_2(s_2'|s_2,a_2)V_{\mathcal{M}_2}^{\pi_2,(h)}(s_2')\right)$$

where we make use of a similar independence property between $p_1(s_1'|s_1,a_1)V_{\mathcal{M}_1}^{\pi_1,(h)}(s_1')$ and $p_2(s_2'|s_2,a_2)$, and the fact that that $p_d(\cdot|s_d,a_d)$ is a probability simplex.

Therefore, we have $Q_{\mathcal{M}}^{\pi,(h+1)}(s,a) = Q_{\mathcal{M}_1}^{\pi_1,(h+1)}(s_1,a_1) + Q_{\mathcal{M}_2}^{\pi_2,(h+1)}(s_2,a_2)$ as desired, where $Q_{\mathcal{M}_d}^{\pi_d,(h+1)}(s_d,a_d) = r_d(s_d,a_d) + \gamma \sum_{s_d'} p_d(s_d'|s_d,a_d)\sum_{a_d'} \pi_j(a_d'|s_d')Q_{\mathcal{M}_j}^{\pi_d,(h)}(s_d',a_d')$.

By mathematical induction, this decomposition holds for any $h$-step Q-function. Letting $h \to \infty$ shows that this holds for the full Q-function. □

## B.2    Sufficient Condition: The Abstraction Setting

We first review some important background on state abstractions. Using the properties of state abstractions, we can prove the main sufficient condition in Theorem 1. This proof follows largely from the techniques used in proving Proposition 7, with the exception of how marginalization over the state space is handled.

***Background on State Abstractions.*** A state abstraction (also known as state aggregation) [65], is a mapping $\phi : \mathcal{S} \to \mathcal{Z}$ that converts each element of the primitive state space $\mathcal{S}$ to an element of the abstract state space $\mathcal{Z}$. Intuitively, if two states $s_1$ and $s_2$ are mapped to the same element under $\phi$, i.e., $\phi(s_1) = \phi(s_2)$, then they are treated as the same (abstract) state under the abstraction. Therefore, we can view an abstraction as a partitioning of the primitive state space into non-overlapping subsets. Since a state abstraction is a many-to-one mapping, we define its inverse as $\phi^{-1}(z) = \{\tilde{s} : \phi(\tilde{s}) = z\}$, a set containing all primitive states that are mapped to the abstract state $z$.

We have the following property of summations involving state abstractions, where for any function $f : \mathcal{S} \to \mathbb{R}$,

$$\sum_{s \in \mathcal{S}} f(s) = \sum_{z \in \mathcal{Z}} \sum_{\tilde{s} \in \phi^{-1}(z)} f(\tilde{s})$$

To understand this property, let us consider the sum of $f(s)$ for all states in $\mathcal{S}$ which can be obtained in two different ways: i) directly iterating through the elements of $\mathcal{S}$, ii) first iterating through the partitions of $\mathcal{S}$ (induced by the abstraction), and then iterating through the elements in each partition, giving rise to the double summation. This property allows us to change the index of summation from primitive states to abstract states. For multiple abstractions $\boldsymbol{\phi} = [\phi_1, \cdots, \phi_D]$ where $\phi_d \neq \phi_{d'}$ if $d \neq d'$, denoting $\boldsymbol{z} = \boldsymbol{\phi}(s) = [z_1, \ldots, z_D]$, we can similarly define the inverse abstraction $\boldsymbol{\phi}^{-1}(\boldsymbol{z}) = \{\tilde{s} : \boldsymbol{\phi}(\tilde{s}) = \boldsymbol{z}\}$, and the summation property similarly applies.

***Proof of Theorem 1.*** Without loss of generality, we consider the setting with $D = 2$ so $\mathcal{A} = \mathcal{A}_1 \times \mathcal{A}_2$; extension to $D > 2$ is straightforward. The proof is based on mathematical induction on a sequence of $h$-step Q-functions of $\pi$ denoted by $Q^{(h)}(s,\boldsymbol{a}) = \mathbb{E}[\sum_{t=1}^h \gamma^{t-1}r_t|s_1 = s, \boldsymbol{a}_1 = \boldsymbol{a}, \boldsymbol{a}_t \sim \pi]$.

***Base case.*** For $h = 1$, the one-step Q-function is simply the reward, which by assumption $r(s,\boldsymbol{a}) = r_1(z_1,a_1) + r_2(z_2,a_2)$. We can trivially set $q_d^{(1)}(z_d,a_d) = r_d(z_d,a_d)$ such that $Q^{(1)}(s,\boldsymbol{a}) = q_1^{(1)}(z_1,a_1) + q_2^{(1)}(z_2,a_2)$.

*Inductive step.* Suppose $Q^{(h)}(s, \boldsymbol{a}) = q_1^{(h)}(z_1, a_1) + q_2^{(h)}(z_2, a_2)$ holds. We can express $Q^{(h+1)}$ in terms of $Q^{(h)}$ using the Bellman equation:

$$Q^{(h+1)}(s, \boldsymbol{a}) = \underbrace{r(s, \boldsymbol{a})}_{①} + \gamma \underbrace{\sum_{s'} p(s'|s, \boldsymbol{a}) V^{(h)}(s')}_{②}$$

where $V^{(h)}(s') = \sum_{\boldsymbol{a}'} \pi(\boldsymbol{a}'|s') Q^{(h)}(s', \boldsymbol{a}')$.

① can be written as a sum $r(s, \boldsymbol{a}) = r_1(z_1, a_1) + r_2(z_2, a_2)$ where each summand depends on only either $a_1$ or $a_2$ but not both. Next we show ② also decomposes in a similar manner.

For a given $s$ we have:

$$V^{(h)}(s) = \sum_{\boldsymbol{a}} \pi(\boldsymbol{a}|s) Q^{(h)}(s, \boldsymbol{a})$$

$$= \sum_{a_1, a_2} \pi_1(a_1|z_1) \pi_2(a_2|z_2) \Big( q_1^{(h)}(z_1, a_1) + q_2^{(h)}(z_2, a_2) \Big)$$

$$= \Big( \underbrace{\sum_{a_2} \pi_2(a_2|z_2)}^{1} \Big) \sum_{a_1} \pi_1(a_1|z_1) q_1^{(h)}(z_1, a_1) + \Big( \underbrace{\sum_{a_1} \pi_1(a_1|z_1)}^{1} \Big) \sum_{a_2} \pi_2(a_2|z_2) q_2^{(h)}(z_2, a_2)$$

$$= \sum_{a_1} \pi_1(a_1|z_1) q_1^{(h)}(z_1, a_1) + \sum_{a_2} \pi_2(a_2|z_2) q_2^{(h)}(z_2, a_2) \, ,$$

where we used the property that $\pi_1(a_1|z_1) q_1^{(h)}(z_1, a_1)$ is independent of $\pi_2(a_2|z_2)$ (and vice versa), and that $\pi_d(\cdot|z_d)$ is a probability simplex. Letting $v_d^{(h)}(z_d) = \sum_{a_d} \pi_d(a_d|z_d) q_d^{(h)}(z_d, a_d)$, then we can write $V^{(h)}(s') = v_1^{(h)}(z_1') + v_2^{(h)}(z_2')$.

Substituting into ②, we have:

$$\sum_{s'} p(s'|s, \boldsymbol{a}) V^{(h)}(s') = \sum_{\boldsymbol{z}'} \sum_{\tilde{s} \in \phi^{-1}(\boldsymbol{z}')} p(\tilde{s}|s, \boldsymbol{a}) V^{(h)}(\tilde{s})$$

$$= \sum_{\boldsymbol{z}'} \sum_{\tilde{s} \in \phi^{-1}(\boldsymbol{z}')} p(\tilde{s}|s, \boldsymbol{a}) V^{(h)}(\tilde{s})$$

$$= \sum_{\boldsymbol{z}'} \Big( \sum_{\tilde{s} \in \phi^{-1}(\boldsymbol{z}')} p(\tilde{s}|s, \boldsymbol{a}) \Big) V^{(h)}(\tilde{s})$$

$$= \sum_{z_1', z_2'} p_1(z_1'|z_1, a_1) p_2(z_2'|z_2, a_2) \Big( v_1^{(h)}(z_1') + v_2^{(h)}(z_2') \Big)$$

$$= \Big( \underbrace{\sum_{z_2'} p_2(z_2'|z_2, a_2)}^{1} \Big) \sum_{z_1'} p_1(z_1'|z_1, a_1) v_1^{(h)}(z_1') + \Big( \underbrace{\sum_{z_1'} p_1(z_1'|z_1, a_1)}^{1} \Big) \sum_{z_2'} p_2(z_2'|z_2, a_2) v_2^{(h)}(z_2')$$

$$= \Big( \sum_{z_1'} p_1(z_1'|z_1, a_1) v_1^{(h)}(z_1') \Big) + \Big( \sum_{z_2'} p_2(z_2'|z_2, a_2) v_2^{(h)}(z_2') \Big)$$

where on the first line we used the property of state abstractions to replace the index of summation, and from the second to the third line we used the fact that for all $\tilde{s} \in \phi^{-1}(\boldsymbol{z}')$ that have the same abstract state vector $\boldsymbol{z}'$, their value $V^{(h)}(s') = v_1^{(h)}(z_1') + v_2^{(h)}(z_2')$ are equal; this allows us to directly sum their transition probabilities $p(\tilde{s}|s, \boldsymbol{a})$. Following that, we substitute in Eqn. (2), and then use a similar independence property as above and that $p_d(\cdot|z_d, a_d)$ is a probability simplex.

Therefore, we have $Q^{(h+1)}(s, \boldsymbol{a}) = q_1^{(h+1)}(z_1, a_1) + q_2^{(h+1)}(z_2, a_2)$ as desired where $q_d^{(h+1)}(z_d, a_d) = r_d(z_d, a_d) + \gamma \sum_{z_d'} p_d(z_d'|z_d, a_d) \sum_{a_d'} \pi(a_d'|z_d') q_d^{(h)}(z_d', a_d')$.

By mathematical induction, this decomposition holds for any $h$-step Q-function. Letting $h \to \infty$ shows that this holds for the full Q-function. □

## B.3 Policy Learning with Bias - Performance Bounds

Consider a particular model-based procedure for approximating the optimal Q-function using Eqn. (1): i) finding approximations $\widehat{\mathcal{M}} = (\hat{p}, \hat{r})$ that are close to the true transition/reward functions $p$, $r$ such that there exists some state abstraction set $\phi$ with $\hat{p}$, $\hat{r}$ satisfying (2) and (3) with respect to $\phi$, ii) doing planning (e.g., dynamic programming) using the approximate MDP parameters $\hat{p}$ and $\hat{r}$. We can show the following performance bounds; note that these upper bounds are loose and information-theoretic (in that they require knowledge of the implicit factorization).

**Proposition 8.** *If the approximation errors in $\hat{p}$ and $\hat{r}$ are upper bounded by $\epsilon_p$ and $\epsilon_r$ for all $s \in \mathcal{S}, \boldsymbol{a} \in \mathcal{A}$:*

$$\sum_{s'} \left| p(s'|s, \boldsymbol{a}) - \hat{p}(s'|s, \boldsymbol{a}) \right| \leq \epsilon_p,$$

$$\left| r(s, \boldsymbol{a}) - \hat{r}(s, \boldsymbol{a}) \right| \leq \epsilon_r,$$

*then the above model-based procedure leads to an approximate Q-function $\hat{Q}$ and an approximate policy $\hat{\pi}$ that satisfy:*

$$\|Q^*_{\mathcal{M}} - Q^*_{\widehat{\mathcal{M}}}\|_\infty \leq \frac{\epsilon_r}{1-\gamma} + \frac{\gamma\epsilon_p R_{\max}}{2(1-\gamma)^2},$$

$$\|V^*_{\mathcal{M}} - V^{\hat{\pi}}_{\mathcal{M}}\|_\infty \leq \frac{2\epsilon_r}{1-\gamma} + \frac{\gamma\epsilon_p R_{\max}}{(1-\gamma)^2}.$$

*Proof.* See classical results by Singh and Yee [66] and Kearns and Singh [67] (the simulation lemma).

## B.4 Subspace of Representable $Q$ Functions

To help understand how the linear parameterization of Q-function Eqn. (1) affects the representation power of the function class, we first define the following matrices for action space featurization.

**Definition 3.** The *sub-action mapping matrix for sub-action space $\mathcal{A}_d$*, $\boldsymbol{\Psi}_d$, is defined as

$$\boldsymbol{\Psi}_j = \begin{bmatrix} - & \boldsymbol{\psi}_d(\boldsymbol{a}^1)^\mathsf{T} & - \\ & \vdots & \\ - & \boldsymbol{\psi}_d(\boldsymbol{a}^{|\mathcal{A}|})^\mathsf{T} & - \end{bmatrix} \in \{0, 1\}^{|\mathcal{A}| \times |\mathcal{A}_d|}$$

where each row $\boldsymbol{\psi}_d(\boldsymbol{a}^i)^\mathsf{T} \in \{0, 1\}^{1 \times |\mathcal{A}_d|}$ is a one-hot vector with a value 1 in column $\text{proj}_{\mathcal{A} \to \mathcal{A}_d}(\boldsymbol{a}^i)$.

*Remark.* The $i$-th row of $\boldsymbol{\Psi}_d$ corresponds to an action $\boldsymbol{a}^i \in \mathcal{A}$, and the $j$-th column corresponds to a particular element of the sub-action space $a^j_d \in \mathcal{A}_d$. The $(i, j)$-entry of $\boldsymbol{\Psi}_d$ is 1 if and only if the projection of $\boldsymbol{a}^i$ onto the sub-action space $\mathcal{A}_d$ is $a^j_d$. Since each row is a one-hot vector, the sum of elements in each row is exactly 1, i.e., $\boldsymbol{\psi}_d(\boldsymbol{a}^i)^\mathsf{T}\mathbf{1} = 1$.

**Definition 4.** The *sub-action mapping matrix*, $\boldsymbol{\Psi}$, is defined by a horizontal concatenation of $\boldsymbol{\Psi}_d$ for $d = 1 \ldots D$

$$\boldsymbol{\Psi} = \begin{bmatrix} \boldsymbol{\Psi}_1 & \cdots & \boldsymbol{\Psi}_D \end{bmatrix} \in \{0, 1\}^{|\mathcal{A}| \times (\sum_d |\mathcal{A}_d|)}$$

*Remark.* $\boldsymbol{\Psi}$ describes how to map each action $\boldsymbol{a}^i \in \mathcal{A}$ to its corresponding sub-actions. Therefore, the sum of elements in each row is exactly $D$, the number of sub-action spaces; $\boldsymbol{\psi}(\boldsymbol{a}^i)^\mathsf{T}\mathbf{1} = D$.

**Definition 5.** The *condensed sub-action mapping matrix*, $\tilde{\boldsymbol{\Psi}}$, is

$$\tilde{\boldsymbol{\Psi}} = \left[ \begin{array}{c|ccc} \mathbf{1} & \tilde{\boldsymbol{\Psi}}_1 & \cdots & \tilde{\boldsymbol{\Psi}}_D \end{array} \right] \in \{0, 1\}^{|\mathcal{A}| \times \left(1 + \sum_d (|\mathcal{A}_d| - 1)\right)}$$

where the first column contains all 1's, and $\tilde{\boldsymbol{\Psi}}_d$ denotes $\boldsymbol{\Psi}_d$ with the first column removed.

**Proposition 9.** $\text{colspace}(\boldsymbol{\Psi}) = \text{colspace}(\tilde{\boldsymbol{\Psi}})$ *and* $\text{rank}(\boldsymbol{\Psi}) = \text{rank}(\tilde{\boldsymbol{\Psi}}) = \text{ncols}(\tilde{\boldsymbol{\Psi}})$ *(i.e., matrix $\tilde{\boldsymbol{\Psi}}$ has full column rank). Consequently, $\boldsymbol{\Psi}\boldsymbol{\Psi}^+ = \tilde{\boldsymbol{\Psi}}\tilde{\boldsymbol{\Psi}}^+$.*

**Corollary 10.** *Suppose the Q-function $Q$ of a policy $\pi$ at state $s$ is linearly decomposable with respect to the sub-actions, i.e., we can write $Q(s, a) = \sum_{d=1}^{D} q_d(s, a_d)$ for all $a_d \in \mathcal{A}_d$, then there exists $\boldsymbol{w}$ and $\tilde{\boldsymbol{w}}$ such that the column vector containing the Q-values for all actions at state $s$ can be expressed as $\boldsymbol{Q}(s, \mathcal{A}) = \boldsymbol{\Psi}\boldsymbol{w} = \tilde{\boldsymbol{\Psi}}\tilde{\boldsymbol{w}}$. In other words, Eqn. (1) is equivalent to $\boldsymbol{Q}(s, \mathcal{A}) \in \text{colspace}(\tilde{\boldsymbol{\Psi}})$.*

**Corollary 11.** *Suppose $\boldsymbol{Q}(s, \mathcal{A}) \notin \text{colspace}(\tilde{\boldsymbol{\Psi}})$. Let $\hat{\boldsymbol{w}} = \boldsymbol{\Psi}^+ \boldsymbol{Q}(s, \mathcal{A})$ and $\hat{\tilde{\boldsymbol{w}}} = \tilde{\boldsymbol{\Psi}}^+ \boldsymbol{Q}(s, \mathcal{A})$ be the least-squares solutions of the respective linear equations. Then $\boldsymbol{\Psi}\hat{\boldsymbol{w}} = \tilde{\boldsymbol{\Psi}}\hat{\tilde{\boldsymbol{w}}}$.*

*Remark.* Corollaries 10 and 11 imply there are two possible implementations, regardless of whether the true Q-function can be represented by the linear parameterization. Intuitively, both versions try to project the true Q-value vector $\boldsymbol{Q}(s, \mathcal{A})$ for a particular state $s$ onto the subspace spanned by the columns of $\boldsymbol{\Psi}$ or $\tilde{\boldsymbol{\Psi}}$. Since the two matrices have the same column space, the results of the projections are equal. This does not imply $\hat{\boldsymbol{w}}$ and $\hat{\tilde{\boldsymbol{w}}}$ are equal (they cannot be as they have different dimensions), but rather the resultant Q-value estimates are equal, $\hat{\boldsymbol{Q}}(s, \mathcal{A}) = \boldsymbol{\Psi}\hat{\boldsymbol{w}} = \tilde{\boldsymbol{\Psi}}\hat{\tilde{\boldsymbol{w}}}$.

To make the theorem statements more concrete, we inspect a simple numerical example and verify the theoretical properties.

**Example 3.** Consider an MDP with $\mathcal{A} = \mathcal{A}_1 \times \mathcal{A}_2$, where $\mathcal{A}_1 = \{0, 1\}$ and $\mathcal{A}_2 = \{0, 1\}$. Consequently, $|\mathcal{A}_1| = |\mathcal{A}_2| = 2$ and $|\mathcal{A}| = 2^2 = 4$.

Suppose for state $s$ we can write $Q(s, a) = Q(s, [a_1, a_2]) = q_1(s, a_1) + q_2(s, a_2)$ for all $a_1 \in \mathcal{A}_1, a_2 \in \mathcal{A}_2$. Then

$$
\boldsymbol{Q}(s, \mathcal{A}) = \begin{bmatrix} Q(s, a_1 = 0, a_2 = 0) \\ Q(s, a_1 = 0, a_2 = 1) \\ Q(s, a_1 = 1, a_2 = 0) \\ Q(s, a_1 = 1, a_2 = 1) \end{bmatrix} = \begin{bmatrix} q_1(s, 0) + q_2(s, 0) \\ q_1(s, 0) + q_2(s, 1) \\ q_1(s, 1) + q_2(s, 0) \\ q_1(s, 1) + q_2(s, 1) \end{bmatrix} = \begin{bmatrix} 1 & 0 & 1 & 0 \\ 1 & 0 & 0 & 1 \\ 0 & 1 & 1 & 0 \\ 0 & 1 & 0 & 1 \end{bmatrix} \begin{bmatrix} q_1(s, 0) \\ q_1(s, 1) \\ q_2(s, 0) \\ q_2(s, 1) \end{bmatrix}
$$

$$
= \begin{bmatrix} | \\ - \boldsymbol{\Psi} - \\ | \end{bmatrix} \begin{bmatrix} | \\ \boldsymbol{w} \\ | \end{bmatrix} \quad \text{where } \boldsymbol{\Psi} = \underbrace{\begin{bmatrix} 1 & 0 \\ 1 & 0 \\ 0 & 1 \\ 0 & 1 \end{bmatrix}}_{\boldsymbol{\Psi}_1} \underbrace{\begin{bmatrix} 1 & 0 \\ 0 & 1 \\ 1 & 0 \\ 0 & 1 \end{bmatrix}}_{\boldsymbol{\Psi}_2}, \quad \boldsymbol{w} = \begin{bmatrix} q_1(s, 0) \\ q_1(s, 1) \\ q_2(s, 0) \\ q_2(s, 1) \end{bmatrix} \begin{matrix} \Big\} \boldsymbol{w}_1 \\ \Big\} \boldsymbol{w}_2 \end{matrix}
$$

We can also write

$$
\boldsymbol{Q}(s, \mathcal{A}) = \tilde{\boldsymbol{\Psi}}\tilde{\boldsymbol{w}}, \quad \text{where } \tilde{\boldsymbol{\Psi}} = \begin{bmatrix} 1 & 0 & 0 \\ 1 & 0 & 1 \\ 1 & 1 & 0 \\ 1 & 1 & 1 \end{bmatrix}, \quad \tilde{\boldsymbol{w}} = \begin{bmatrix} v_0(s) \\ u_1(s) \\ u_2(s) \end{bmatrix} = \begin{bmatrix} q_1(s, 0) + q_2(s, 0) \\ q_1(s, 1) - q_1(s, 0) \\ q_2(s, 1) - q_2(s, 0) \end{bmatrix}
$$

One can verify that $\text{rank}(\boldsymbol{\Psi}) = \text{rank}(\tilde{\boldsymbol{\Psi}}) = 3$ and $\text{colspace}(\boldsymbol{\Psi}) = \text{colspace}(\tilde{\boldsymbol{\Psi}})$, because the columns of $\tilde{\boldsymbol{\Psi}}$ are linearly independent, but the columns of $\boldsymbol{\Psi}$ are not linearly independent:

$$
\begin{bmatrix} 1 \\ 1 \\ 0 \\ 0 \end{bmatrix} + \begin{bmatrix} 0 \\ 0 \\ 1 \\ 1 \end{bmatrix} - \begin{bmatrix} 1 \\ 0 \\ 1 \\ 0 \end{bmatrix} = \begin{bmatrix} 0 \\ 1 \\ 0 \\ 1 \end{bmatrix}.
$$

Furthermore,

$$
\boldsymbol{\Psi}^+ = \begin{bmatrix} 3/8 & 3/8 & -1/8 & -1/8 \\ -1/8 & -1/8 & 3/8 & 3/8 \\ 3/8 & -1/8 & 3/8 & -1/8 \\ -1/8 & 3/8 & -1/8 & 3/8 \end{bmatrix}, \quad \tilde{\boldsymbol{\Psi}}^+ = \begin{bmatrix} 3/4 & 1/4 & 1/4 & -1/4 \\ -1/2 & -1/2 & 1/2 & 1/2 \\ -1/2 & 1/2 & -1/2 & 1/2 \end{bmatrix}
$$

and

$$
\boldsymbol{\Psi}\boldsymbol{\Psi}^+ = \tilde{\boldsymbol{\Psi}}\tilde{\boldsymbol{\Psi}}^+ = \begin{bmatrix} 3/4 & 1/4 & 1/4 & -1/4 \\ 1/4 & 3/4 & -1/4 & 1/4 \\ 1/4 & -1/4 & 3/4 & 1/4 \\ -1/4 & 1/4 & 1/4 & 3/4 \end{bmatrix}.
$$

◁

*Proof of Proposition 9.*

First note that $\boldsymbol{\Psi}$ is a tall matrix for non-trivial cases, with more rows than columns, because $|\mathcal{A}| = \prod_d |\mathcal{A}_d| \geq \sum_d |\mathcal{A}_d|$ if $|\mathcal{A}_d| \geq 2$ for all $d$ (see proof). Therefore, the rank of $\boldsymbol{\Psi}$ is the number of linear independent columns of $\boldsymbol{\Psi}$.

We use the following notation to write matrix $\boldsymbol{\Psi}_d$ in terms of its columns:

$$\boldsymbol{\Psi}_d = \begin{bmatrix} | & & | \\ \boldsymbol{c}_{d,1} & \cdots & \boldsymbol{c}_{d,|\mathcal{A}_d|} \\ | & & | \end{bmatrix}.$$

The following statements are true:

**Claim 1:** The columns of $\boldsymbol{\Psi}_d$ are pairwise orthogonal, $\boldsymbol{c}_{d,j}{}^{\mathsf{T}} \boldsymbol{c}_{d,j'} = 0, \forall j \neq j'$, and they form an orthogonal basis. This is because each row $\boldsymbol{\psi}_d(\boldsymbol{a}^i)^{\mathsf{T}}$ is a one-hot vector, containing only one 1; this implies that out of the two entries in row $i$ of $\boldsymbol{c}_{d,j}$ and $\boldsymbol{c}_{d,j'}$, at least one entry is 0, and their product must be 0.

**Claim 2:** The sum of entries in each row of $\boldsymbol{\Psi}_d$ is 1, and $\sum_{j=1}^{|\mathcal{A}_d|} \boldsymbol{c}_{d,j} = \mathbf{1}$ a column vector of 1's with matching size. This is a direct consequence of each row $\boldsymbol{\psi}_d(\boldsymbol{a}^i)^{\mathsf{T}}$ being a one-hot vector. In other words, $\mathbf{1} \in \text{colspace}(\boldsymbol{\Psi}_d)$.

**Claim 3:** The columns of $\boldsymbol{\Psi}$ are not linearly independent. This is because there is not a unique way to write $\mathbf{1}$ as a linear combination of the columns of $\boldsymbol{\Psi}$. For example, $\sum_{j=1}^{|\mathcal{A}_d|} \boldsymbol{c}_{d,j} = \sum_{j=1}^{|\mathcal{A}_{d'}|} \boldsymbol{c}_{d',j} = \mathbf{1}$ for some $d' \neq d$, where we used the columns of $\boldsymbol{\Psi}_d$ and $\boldsymbol{\Psi}_{d'}$.

**Claim 4:** $\mathbf{1} \notin \text{colspace}(\tilde{\boldsymbol{\Psi}}_1 \cdots \tilde{\boldsymbol{\Psi}}_D)$ because the first entry of every column vector in any $\tilde{\boldsymbol{\Psi}}_d$ is 0 and no linear combination of them can result in a 1. Consequently, $\mathbf{1} \notin \text{colspace}(\tilde{\boldsymbol{\Psi}}_d)$ for any $d$.

**Claim 5:** $\boldsymbol{c}_{d,1} \notin \text{colspace}(\mathbf{1}, \tilde{\boldsymbol{\Psi}}_{d'} : d' \neq d)$, where $\boldsymbol{c}_{d,1}$ is the column removed from $\boldsymbol{\Psi}_d$ to construct $\tilde{\boldsymbol{\Psi}}_d$. This can also be seen from the first entry of the column vector: the first entry of $\boldsymbol{c}_{d,1}$ is 1, and all columns of $\tilde{\boldsymbol{\Psi}}_{d'} : d' \neq d$ have the first entry being 0.

**Claim 6:** $\boldsymbol{c}_{d,j} \notin \text{colspace}(\mathbf{1}, \tilde{\boldsymbol{\Psi}}_1 \cdots \tilde{\boldsymbol{\Psi}}_D \setminus \{\boldsymbol{c}_{d,j}\})$ for $j > 1$. By expressing $\boldsymbol{c}_{d,j} = (\mathbf{1} - \sum_{j'=2, j' \neq j}^{|\mathcal{A}_d|} \boldsymbol{c}_{d,j'}) + (-\boldsymbol{c}_{d,1})$, we observe that the first part of the sum lies in the column space, while the second part does not (from the previous claim, $\boldsymbol{c}_{d,1}$ is not in the column space of $\tilde{\boldsymbol{\Psi}}_{d'}$ where $d' \neq d$; this is because within $\tilde{\boldsymbol{\Psi}}_d$, the only way is $\boldsymbol{c}_{d,1} = \mathbf{1} - \sum_{j'=2}^{|\mathcal{A}_d|} \boldsymbol{c}_{d,j'}$ and we have excluded one of the columns $\boldsymbol{c}_{d,j}$ from the column space).

Combining these claims implies that each column of $\tilde{\boldsymbol{\Psi}}$ cannot be expressed as a linear combination of all other columns, and thus $\tilde{\boldsymbol{\Psi}}$ has full column rank, $\text{rank}(\tilde{\boldsymbol{\Psi}}) = \text{ncols}(\tilde{\boldsymbol{\Psi}}) = 1 + \sum_{d=1}^{D}(|\mathcal{A}_d| - 1)$. It follows that $\tilde{\boldsymbol{\Psi}}$ contains the linearly independent subset of columns from $\boldsymbol{\Psi}$, and their column spaces and ranks are equal.

$\boldsymbol{\Psi}\boldsymbol{\Psi}^+$ and $\tilde{\boldsymbol{\Psi}}\tilde{\boldsymbol{\Psi}}^+$ are orthogonal projection matrices onto the column space of $\boldsymbol{\Psi}$ and $\tilde{\boldsymbol{\Psi}}$, respectively. Since $\text{colspace}(\boldsymbol{\Phi}) = \text{colspace}(\tilde{\boldsymbol{\Psi}})$, it follows that $\boldsymbol{\Psi}\boldsymbol{\Psi}^+ = \tilde{\boldsymbol{\Psi}}\tilde{\boldsymbol{\Psi}}^+$. $\qquad\square$

## B.5 A Necessary Condition for Unbiasedness

Consider the matrix form of the Bellman equation (cf. Sec 2 of Lagoudakis and Parr [68]):

$$\boldsymbol{Q} = \boldsymbol{R} + \gamma \boldsymbol{P}^\pi \boldsymbol{Q}$$

where $\boldsymbol{Q} \in \mathbb{R}^{|\mathcal{S}||\mathcal{A}|}$ is a vector containing the Q-values for all state-action pairs, $\boldsymbol{R} \in \mathbb{R}^{|\mathcal{S}||\mathcal{A}|}$, and $\boldsymbol{P}^\pi \in \mathbb{R}^{|\mathcal{S}||\mathcal{A}| \times |\mathcal{S}||\mathcal{A}|}$ is the $(s, a)$-transition matrix induced by the MDP and policy $\pi$. Solving this

equation gives us the Q-function in closed form:

$$\boldsymbol{Q} = (\boldsymbol{I} - \gamma \boldsymbol{P}^\pi)^{-1} \boldsymbol{R} \tag{5}$$

where $\boldsymbol{I} \in \mathbb{R}^{|\mathcal{S}||\mathcal{A}| \times |\mathcal{S}||\mathcal{A}|}$.

To derive a necessary condition, we start by assuming that the Q-function is representable by the linear parameterization, i.e., there exists $\boldsymbol{W} \in \mathbb{R}^{(\sum_{d=1}^D |\mathcal{A}_d|) \times |\mathcal{S}|}$ such that $\mathrm{vec}^{-1}_{|\mathcal{A}| \times |\mathcal{S}|}(\boldsymbol{Q}) = \boldsymbol{\Psi} \boldsymbol{W}$. Here, $\mathrm{vec}^{-1}_{|\mathcal{A}| \times |\mathcal{S}|}$ is the inverse vectorization operator that reshapes the vector of all Q-values into a matrix of size $|\mathcal{A}| \times |\mathcal{S}|$, and $\boldsymbol{\Psi} \in \{0,1\}^{|\mathcal{A}| \times (\sum_{d=1}^D |\mathcal{A}_d|)}$ is defined in Appendix B.4. Substituting Eqn. (5) into the premise gives us a necessary condition: if there exists $\boldsymbol{W} \in \mathbb{R}^{(\sum_{d=1}^D |\mathcal{A}_d|) \times |\mathcal{S}|}$ such that

$$\mathrm{vec}^{-1}_{|\mathcal{A}| \times |\mathcal{S}|}\left( (\boldsymbol{I} - \gamma \boldsymbol{P}^\pi)^{-1} \boldsymbol{R} \right) = \boldsymbol{\Psi} \boldsymbol{W}$$

Unfortunately, unlike the sufficient conditions in Theorem 1 (and Proposition 7), this necessary condition is not as clean and likely not verifiable in most settings. The matrix inverse and $\mathrm{vec}^{-1}$ reshaping operation make it challenging to further manipulate the expression. This highlights the non-trivial nature of the problem.

## B.6 Variance Reduction in the Bandit Setting

***Background on Rademacher complexity.*** Let $\mathcal{F}$ be a family of functions mapping from $\mathbb{R}^d$ to $\mathbb{R}$. The empirical Rademacher complexity of $\mathcal{F}$ for a sample $\mathcal{S} = \{\mathbf{x}_1, \dots, \mathbf{x}_m\}$ is defined by

$$\widehat{\mathfrak{R}}_{\mathcal{S}}(\mathcal{F}) = \mathbb{E}_{\boldsymbol{\sigma}} \left[ \sup_{f \in \mathcal{F}} \frac{1}{m} \sum_{i=1}^m \sigma_i f(\mathbf{x}_i) \right],$$

where $\boldsymbol{\sigma} = [\sigma_1, \dots, \sigma_m]$ is a vector of i.i.d. Rademacher variables, i.e., independent uniform r.v.s taking values in $\{-1, +1\}$.

For a matrix $\mathbf{M} \in \mathbb{R}^{m \times D}$, define the $(p, q)$-group norm as the $q$-norm of the $p$-norm of the columns of $\mathbf{M}$, that is $\|\mathbf{M}\|_{p,q} = \|[\|\mathbf{M}_1\|_p, \cdots, \|\mathbf{M}_D\|_p]\|_q$, where $\mathbf{M}_j$ is the $j$-th column of $\mathbf{M}$.

In Awasthi et al. [69], Theorem 2 stated that: let $\mathcal{F} = \{f = \mathbf{w}^\mathsf{T}\mathbf{x} : \|\mathbf{w}\|_p \leq A\}$ be a family of linear functions defined over $\mathbb{R}^d$ with bounded weight in $\ell_2$-norm, then the empirical Rademacher complexity of $\mathcal{F}$ for a sample $\mathcal{S} = \{\mathbf{x}_1, \dots, \mathbf{x}_m\}$ satisfies the following lower bound (where $\mathbf{X} = [\mathbf{x}_1 \dots \mathbf{x}_m]^\mathsf{T}$):

$$\widehat{\mathfrak{R}}_{\mathcal{S}}(\mathcal{F}) \geq \frac{A}{\sqrt{2}m} \|\mathbf{X}\|_{2,2}.$$

*Proof for Proposition 5.* For the sake of argument, we consider the one-timestep bandit setting; extension to the sequential setting can be similarly derived following Chen and Jiang [62], Duan et al. [17]. Let the true generative model be $\boldsymbol{Q}^* = \boldsymbol{\Psi}\boldsymbol{r} + \boldsymbol{\psi}_{\mathrm{Interact}} r_{\mathrm{Interact}}$ (details in Appendix B.8). We formally show the reduction in the variance of the estimators, by comparing the lower bound of their respective empirical Rademacher complexities. A smaller Rademacher complexity translates into lower variance estimators.

Suppose we obtain a sample of $m$ actions and apply the linear approximation. Our approach for factored action space corresponds to the matrix $\boldsymbol{X} \in \{0,1\}^{m \times (\sum_d |\mathcal{A}_d|)}$, obtained by stacking the corresponding rows of $\boldsymbol{\Psi}$ (recall Definition 4). The complete, combinatorial action space corresponds to the matrix $\boldsymbol{X}' = [\boldsymbol{X}, \boldsymbol{x}_{\mathrm{Interact}}] \in \{0,1\}^{m \times (1+\sum_d |\mathcal{A}_d|)}$ by adding the corresponding rows of $\boldsymbol{\psi}_{\mathrm{Interact}}$. By definition, $\|\boldsymbol{X}\|_{p,q} < \|\boldsymbol{X}'\|_{p,q}$, since the former drops a column with non-zero norm that exists in the latter.

Consider the following two function families, for the factored action space and the complete action space respectively:

$$\mathcal{F}_{\mathrm{F}} = \{f = \mathbf{w}_{\mathrm{F}}^\mathsf{T}\mathbf{x} : \|\mathbf{w}_{\mathrm{F}}\|_2 \leq A\}$$
$$\mathcal{F}_{\mathrm{C}} = \{f = \mathbf{w}_{\mathrm{C}}^\mathsf{T}\mathbf{x}' : \|\mathbf{w}_{\mathrm{C}}\|_2 \leq A\},$$

for some $A > 0$. A straightforward application of Theorem 2 of Awasthi et al. [69] shows that the lower bound on the Rademacher complexity of the of the factored action space is smaller than that of the complete action space, which completes our argument. □

## B.7 Standardization of Rewards for the Bandit Setting (Proposition 6)

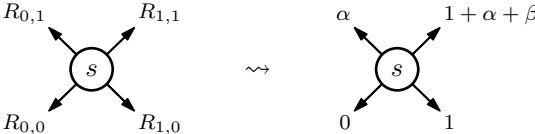

Figure 9: Standardization of rewards.

Suppose the rewards of the four arms are $[R_{0,0}, R_{0,1}, R_{1,0}, R_{1,1}]$. We can apply the following transformations to reduce any reward function to the form of $[0, \alpha, 1, 1+\alpha+\beta]$, and these transformations do not affect the least-squares solution:

- If $R_{0,0} = R_{1,0}$ and $R_{0,1} = R_{1,1}$, we can ignore x-axis sub-action as setting it to either $0$ ($\leftarrow$) or $1$ ($\rightarrow$) does not affect the reward. Similarly, if $R_{0,0} = R_{0,1}$ and $R_{1,0} = R_{1,1}$, we can ignore y-axis sub-action. In both cases, this reduces to a one-dimensional action space which we do not discuss further.

- Now at least one of the following is false: $R_{0,0} = R_{1,0}$ or $R_{0,1} = R_{1,1}$. If $R_{0,0} \neq R_{1,0}$, skip this step. Otherwise, it must be that $R_{0,0} = R_{1,0}$ and $R_{0,1} \neq R_{1,1}$. Swap the role of down vs. up such that the new $R_{0,0} \neq R_{1,0}$.

- If $R_{0,0} < R_{1,0}$, skip this step. Otherwise it must be that $R_{0,0} > R_{1,0}$. Swap the role of left vs. right so that $R_{0,0} < R_{1,0}$.

- If $R_{0,0} \neq 0$, subtract $R_{0,0}$ from all rewards so that the new $R_{0,0} = 0$.

- Now $R_{1,0} > R_{0,0} > 0$ must be positive. If $R_{1,0} \neq 1$, divide all rewards by $R_{1,0}$ so that the new $R_{1,0} = 1$.

- Lastly, we should have $R_{0,0} = 0$ and $R_{1,0} = 1$. Set $\alpha = R_{0,1}$ and $\beta = R_{1,1} - R_{1,0} - R_{0,1}$.

## B.8 Omitted-Variable Bias in the Bandit Setting (Proposition 6)

Suppose the true generative model is

$$Q^*(\boldsymbol{a}) = \mathbb{1}_{(a_x=\text{Left})} r_{\text{Left}} + \mathbb{1}_{(a_x=\text{Right})} r_{\text{Right}} + \mathbb{1}_{(a_y=\text{Down})} r_{\text{Down}} + \mathbb{1}_{(a_y=\text{Up})} r_{\text{Up}} + \mathbb{1}_{(\boldsymbol{a}=\text{Right,Up})} r_{\text{Interact}}$$

In other words,

$$\begin{bmatrix} Q^*(\swarrow) \\ Q^*(\nwarrow) \\ Q^*(\searrow) \\ Q^*(\nearrow) \end{bmatrix} = \begin{bmatrix} 1 & 0 & 1 & 0 \\ 1 & 0 & 0 & 1 \\ 0 & 1 & 1 & 0 \\ 0 & 1 & 0 & 1 \end{bmatrix} \begin{bmatrix} r_{\text{Left}} \\ r_{\text{Right}} \\ r_{\text{Down}} \\ r_{\text{Up}} \end{bmatrix} + \begin{bmatrix} 0 \\ 0 \\ 0 \\ 1 \end{bmatrix} r_{\text{Interact}} \quad \rightsquigarrow \quad \boldsymbol{Q}^* = \boldsymbol{\Psi} \boldsymbol{r} + \boldsymbol{\psi}_{\text{Interact}} r_{\text{Interact}}$$

Here, $r_{\text{Left}}, r_{\text{Right}}, r_{\text{Down}}, r_{\text{Up}}, r_{\text{Interact}}$ are parameters of the generative model. Note that the matrix $[\boldsymbol{\Psi}, \boldsymbol{\psi}_{\text{Interact}}]$ has a column space of $\mathbb{R}^4$, i.e., this generative model captures every possible reward configuration of the four actions.

Applying our proposed linear approximation translates to "dropping" the interaction parameter, $r_{\text{Interact}}$, and estimate the remaining four parameters. This leads to a form of omitted-variable bias, which can be computed as:

$$\boldsymbol{\Psi}^+ \boldsymbol{\psi}_{\text{Interact}} r_{\text{Interact}} = \begin{bmatrix} 1 & 0 & 1 & 0 \\ 1 & 0 & 0 & 1 \\ 0 & 1 & 1 & 0 \\ 0 & 1 & 0 & 1 \end{bmatrix}^+ \begin{bmatrix} 0 \\ 0 \\ 0 \\ 1 \end{bmatrix} r_{\text{Interact}}$$

$$= \begin{bmatrix} 3/8 & 3/8 & -1/8 & -1/8 \\ -1/8 & -1/8 & 3/8 & 3/8 \\ 3/8 & -1/8 & 3/8 & -1/8 \\ -1/8 & 3/8 & -1/8 & 3/8 \end{bmatrix} \begin{bmatrix} 0 \\ 0 \\ 0 \\ 1 \end{bmatrix} r_{\text{Interact}} = \begin{bmatrix} -1/8 \\ 3/8 \\ -1/8 \\ 3/8 \end{bmatrix} r_{\text{Interact}}$$

The biased estimate of the four parameters are:

$$\hat{\boldsymbol{r}} = \boldsymbol{r} + \boldsymbol{\Psi}^{+}\boldsymbol{\psi}_{\text{Interact}} r_{\text{Interact}} \quad \rightsquigarrow \quad \begin{bmatrix} \hat{r}_{\text{Left}} \\ \hat{r}_{\text{Right}} \\ \hat{r}_{\text{Down}} \\ \hat{r}_{\text{Up}} \end{bmatrix} = \begin{bmatrix} r_{\text{Left}} - \frac{1}{8}r_{\text{Interact}} \\ r_{\text{Right}} + \frac{3}{8}r_{\text{Interact}} \\ r_{\text{Down}} - \frac{1}{8}r_{\text{Interact}} \\ r_{\text{Up}} + \frac{3}{8}r_{\text{Interact}} \end{bmatrix}$$

and the estimated Q-values are:

$$\hat{\boldsymbol{Q}} = \begin{bmatrix} \hat{Q}(\swarrow) \\ \hat{Q}(\nwarrow) \\ \hat{Q}(\searrow) \\ \hat{Q}(\nearrow) \end{bmatrix} = \begin{bmatrix} 1 & 0 & 1 & 0 \\ 1 & 0 & 0 & 1 \\ 0 & 1 & 1 & 0 \\ 0 & 1 & 0 & 1 \end{bmatrix} \begin{bmatrix} r_{\text{Left}} - \frac{1}{8}r_{\text{Interact}} \\ r_{\text{Right}} + \frac{3}{8}r_{\text{Interact}} \\ r_{\text{Down}} - \frac{1}{8}r_{\text{Interact}} \\ r_{\text{Up}} + \frac{3}{8}r_{\text{Interact}} \end{bmatrix} = \begin{bmatrix} r_{\text{Left}} + r_{\text{Down}} - \frac{1}{4}r_{\text{Interact}} \\ r_{\text{Left}} + r_{\text{Up}} + \frac{1}{4}r_{\text{Interact}} \\ r_{\text{Right}} + r_{\text{Down}} + \frac{1}{4}r_{\text{Interact}} \\ r_{\text{Right}} + r_{\text{Up}} + \frac{3}{4}r_{\text{Interact}} \end{bmatrix}$$

For the bandit problem in Figure 4a, substituting $r_{\text{Left}} + r_{\text{Down}} = 0$, $r_{\text{Left}} + r_{\text{Up}} = \alpha$, $r_{\text{Right}} + r_{\text{Down}} = 1$, and $r_{\text{Interact}} = \beta$ gives

$$\begin{bmatrix} \hat{Q}(\swarrow) \\ \hat{Q}(\nwarrow) \\ \hat{Q}(\searrow) \\ \hat{Q}(\nearrow) \end{bmatrix} = \begin{bmatrix} -\frac{1}{4}\beta \\ \alpha + \frac{1}{4}\beta \\ 1 + \frac{1}{4}\beta \\ 1 + \alpha + \frac{3}{4}\beta \end{bmatrix}$$

which is the solution we presented in Figure 4c.

### B.9 Accounting for Sub-action Interactions

When the interaction effect is not negligible and can lead to suboptimal performance, one solution is to explicitly encode the residual interaction terms in the decomposed Q-function by letting $Q(s, \boldsymbol{a}) = \sum_{d=1}^{D} q_d(s, a_d) + \Re(\boldsymbol{a})$. The exact parameterization of the residual term $\Re(\boldsymbol{a})$ is problem dependent: one may incorporate Tavakoli et al. [6] to systematically consider interactions of certain "ranks" (e.g., limiting it to only two-way or three-way interactions), and consider regularizing the magnitude of residual terms so we still benefit from the efficiency gains of the linear decomposition.

## C More Illustrative Examples

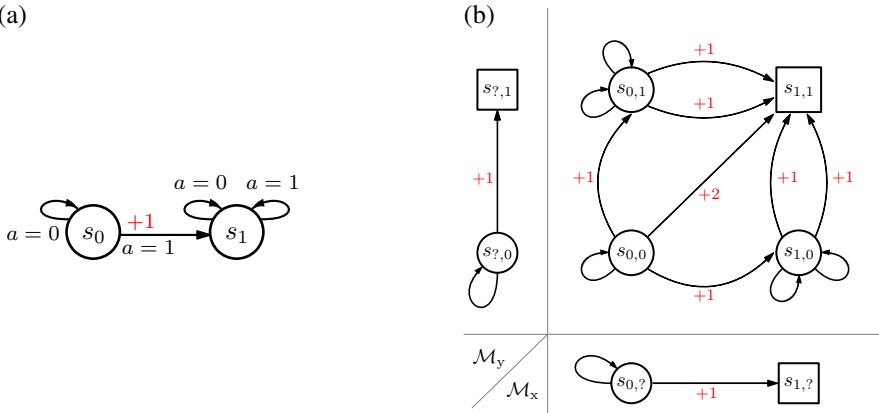

Figure 10: (a) A one-dimensional chain MDP, with an initial state $s_0$ and an absorbing state $s_1$, and two actions $a = 0$ (left) and $a = 1$ (right). (b) A two-dimensional chain MDP shown together with the component chains $\mathcal{M}_{\text{x}}$ and $\mathcal{M}_{\text{y}}$. Rewards are denoted in red. Squares □ indicate absorbing states whose outgoing transition arrows are omitted. For readability, in the diagram, the states and actions are laid out following a convention similar to the Cartesian coordinate system so that the bottom left state has index $(0, 0)$, and right and up both increase the corresponding coordinate by 1.

In this appendix, we discuss the building blocks of the examples used in the main paper and provide additional examples to support the theoretical properties presented in Section 3.

**One-dimensional Chain.** First, consider the chain problem depicted in Figure 10a. The agent always starts in the initial state $s_0$ and can take one of two possible actions: left ($a = 0$), which leads the agent to stay at $s_0$, or right ($a = 1$), which leads the agent to transition into $s_1$ and receive a reward of $+1$. After reaching the absorbing state $s_1$, both $a = 0$ and $a = 1$ lead the agent to stay at $s_1$ with zero reward. For $\gamma < 1$, a (deterministic) optimal policy is $\pi^*(s_0) = 1$, and either action can be taken in $s_1$. Next, we use this MDP to construct a two-dimensional problem.

**Two-dimensional Chain.** Following the construction used in Definition 1, we consider an MDP $\mathcal{M} = \mathcal{M}_x \times \mathcal{M}_y$ consisting of two chains (the horizontal chain $\mathcal{M}_x$ and the vertical chain $\mathcal{M}_y$) running in parallel, as shown in Figure 10b. Their corresponding state spaces are $\mathcal{S}_x = \{s_{0,?}, s_{1,?}\}$ and $\mathcal{S}_y = \{s_{?,0}, s_{?,1}\}$, which indicate the x- and y-coordinates respectively. There are 4 actions from each state, depicted by diagonal arrows $\{\swarrow, \nwarrow, \searrow, \nearrow\}$; each action $\boldsymbol{a} = [a_x, a_y]$ effectively leads the agent to perform $a_x$ in $\mathcal{M}_x$ and $a_y$ in $\mathcal{M}_y$. For example, taking action $\nearrow = [\rightarrow, \uparrow]$ from state $s_{0,0}$ leads the agent to transition into state $s_{1,1}$ and receive a reward of $+2$ (the sum of $+1$ from $\mathcal{M}_x$ and $+1$ from $\mathcal{M}_y$). For $\gamma < 1$, an optimal policy for this MDP is to always move up and right, $\pi^*(\cdot) = \nearrow = [\rightarrow, \uparrow]$, regardless of which state the agent is in.

**Satisfying the Sufficient Conditions.** Let $\phi_x : \mathcal{S} \rightarrow \mathcal{S}_x$ and $\phi_y : \mathcal{S} \rightarrow \mathcal{S}_y$ be the abstractions. By construction, the transition and reward functions of this MDP satisfy Eqns. (3) and (4). To apply Theorem 1, the policy must satisfy Eqn. (4). In Figure 11, we show three such policies (other policies in this category are omitted due to symmetry and transitions that have the same outcome), together with the true Q-functions (with $\gamma = 0.9$) and their decompositions in the form of Eqn. (1).

**Violating the Sufficient Conditions.**

- **Policy violates Eqn. (4) - Nonzero bias.** For this setting, we hold the MDP (transitions and rewards) unchanged. In Figure 12, we show seven policies that do not satisfy Eqn. (4), together with the resultant Q-function and the biased linear approximation with the non-zero approximation error.

- **Transition violates Eqn. (2) - Nonzero Bias.** Figure 13 shows an example where one transition has been modified.

- **Reward violates Eqn. (2) - Nonzero Bias.** Figure 14 shows an example where one reward has been modified.

- **Transition violates Eqn. (2), or policy violates Eqn. (4) - Zero Bias.** If $\gamma = 0$, then the Q-function is simply the immediate reward, and any conditions on the transition or policy can be forgone.

- **Reward violates Eqn. (3) - Zero Bias.** It is possible to construct reward functions adversarially such that $r$ itself does not satisfy the condition, and yet $Q$ can be linearly decomposed. See Figure 15 for an example.

Figure 11: Example MDPs and policies where Proposition 7 applies, for the optimal policy and two particular non-optimal policies. $\gamma = 0.9$. We show the linear decomposition of the Q-function into $Q_x$ and $Q_y$. $Q_x$ only depends on the x-coordinate of state and the sub-action that moves $\leftarrow$ or $\rightarrow$; $Q_y$ only depends on the y-coordinate of state and the sub-action that moves $\downarrow$ or $\uparrow$.

Figure 12: Example MDPs and policies where Proposition 7 does not apply because the policy violates Eqn. (4) (violations are highlighted). $\gamma = 0.9$. For example, in the first case, the policy does not take the same sub-action from $s_{0,0}$ and $s_{0,1}$ with respect to the horizontal chain $\mathcal{M}_x$. Applying the linear approximation produces biased estimates $\hat{Q}$ of the true Q-function, $Q^\pi$.

| $\pi(\mathcal{S})$ | MDP diagram | $Q^\pi(s_{0,0}, \mathcal{A})$ | $\hat{Q}(s_{0,0}, \mathcal{A})$ |
|---|---|---|---|

$$s_{0,0}\begin{bmatrix}\nearrow\\\nearrow\\\nearrow\\\nearrow\end{bmatrix} = \begin{bmatrix}\rightarrow,\uparrow\\\rightarrow,\uparrow\\\rightarrow,\uparrow\\\rightarrow,\uparrow\end{bmatrix} \qquad \begin{bmatrix}1.8\\1.9\\1\\2\end{bmatrix} \qquad \begin{bmatrix}1.575\\2.125\\1.225\\1.775\end{bmatrix}$$

Figure 13: Example MDPs and policies where Theorem 1 does not apply because the transition function violates Eqn. (2). $\gamma = 0.9$. In this example, the highlighted transition corresponding to the action $\nearrow = [\rightarrow, \uparrow]$ from $s_{0,1}$ does not move right ($\rightarrow$ under $\mathcal{M}_x$) to $s_{1,1}$ and instead moves back to state $s_{0,1}$. Applying the linear approximation produces biased estimates $\hat{Q}$ of the true Q-function, $Q^\pi$.

| Reward function | Q-function | $Q^\pi(s_{0,0}, \mathcal{A})$ | $\hat{Q}(s_{0,0}, \mathcal{A})$ |
|---|---|---|---|

$$\begin{bmatrix}0.9\\1.9\\1.9\\1\end{bmatrix} \qquad \begin{bmatrix}1.375\\1.425\\1.425\\1.475\end{bmatrix}$$

Figure 14: Example MDPs and policies where Theorem 1 does not apply because the reward function violates Eqn. (3). $\gamma = 0.9$. In this example, the reward function of the bottom left state $s_{0,0}$ does not satisfy the condition because the reward of $\nearrow$ is $1 \neq 2 = 1 + 1$. Applying the linear approximation produces biased estimates $\hat{Q}$ of the true Q-function, $Q^\pi$.

| Reward function | Q-function | | $Q^\pi$ | | | | = | | $Q_x$ | | | + | | $Q_y$ | | |
|---|---|---|---|---|---|---|---|---|---|---|---|---|---|---|---|---|

|  |  | $\swarrow$ | $\nwarrow$ | $\searrow$ | $\nearrow$ |  |  | $\leftarrow$ | $\leftarrow$ | $\rightarrow$ | $\rightarrow$ |  |  | $\downarrow$ | $\uparrow$ | $\downarrow$ | $\uparrow$ |
|---|---|---|---|---|---|---|---|---|---|---|---|---|---|---|---|---|---|
| $s_{0,0}$ | | 8.5 | 3 | 7 | 1.5 | | $s_{0,?}$ | 1.5 | 1.5 | 0 | 0 | | $s_{?,0}$ | 7 | 1.5 | 7 | 1.5 |
| $s_{0,1}$ | | 0 | 0 | 1 | 1 | | $s_{0,?}$ | 0 | 0 | 1 | 1 | | $s_{?,1}$ | 0 | 0 | 0 | 0 |
| $s_{1,0}$ | | 0 | 4 | 0 | 4 | | $s_{1,?}$ | 0 | 0 | 0 | 0 | | $s_{?,0}$ | 0 | 4 | 0 | 4 |
| $s_{1,1}$ | | 0 | 0 | 0 | 0 | | $s_{1,?}$ | 0 | 0 | 0 | 0 | | $s_{?,1}$ | 0 | 0 | 0 | 0 |

Figure 15: Example MDPs and policies where Theorem 1 does not apply because the reward function violates Eqn. (3). $\gamma = 1$. In this example, the reward function of the bottom left state $s_{0,0}$ does not satisfy the condition because $7 + 1.5 \neq 2 + 3$. However, there exists a linear decomposition of the true Q-function, $Q^\pi$, for a particular policy denoted by bold blue arrows.

# D Experiments

## D.1 Sepsis Simulator - Implementation Details

When generating the datasets, we follow the default initial state distribution specified in the original implementation.

By default, we used neural networks consisting of one hidden layer with 1,000 neurons and ReLU activation to allow for function approximators with sufficient expressivity. We trained these networks using the Adam optimizer (default settings) [70] with a batch size of 64 for a maximum of 100 epochs, applying early stopping on 10% "validation data" (specific to each supervised task) with a patience of 10 epochs. We minimized the mean squared error (MSE) for regression tasks (each iteration of FQI). For FQI, we also added value clipping (to be within the range of possible returns $[-1, 1]$) when computing bootstrapping targets to ensure a bounded function class and encourage better convergence behavior [71].

## D.2 MIMIC Sepsis - Implementation Details

The RNN AIS encoder was trained to predict the mean of a unit-variance multivariate Gaussian that outputs the observation at subsequent timesteps, conditioned on the subsequent actions, following the idea in Subramanian and Mahajan [37]. We performed a grid search over the hyperparameters (Table 2) for training the RNN, selecting the model that achieved the smallest validation loss. Using the best encoder model, we then trained the offline RL policy using BCQ (and factored BCQ), considering validation performance of all checkpoints (saved every 100 iterations, for a maximum of 10,000 iterations) and all combinations of the BCQ hyperparameters (Table 2).

Table 2: Hyperparameter values used for training the RNN approximate information state as well as BCQ for offline RL. Discrete BCQ for both the baseline and factored implementation are identical except for the final layer of the Q-networks.

| Hyperparameter | Searched Settings |
|---|---|
| RNN: | |
| - Embedding dimension, $d_S$ | $\{8, 16, 32, 64, 128\}$ |
| - Learning rate | $\{$ 1e-5, 5e-4, 1e-4, 5e-3, 1e-3 $\}$ |
| BCQ (with 5 random restarts): | |
| - Threshold, $\tau$ | $\{0, 0.01, 0.05, 0.1, 0.3, 0.5, 0.75, 0.999\}$ |
| - Learning rate | 3e-4 |
| - Weight decay | 1e-3 |
| - Hidden layer size | 256 |

## D.3 MIMIC Sepsis results

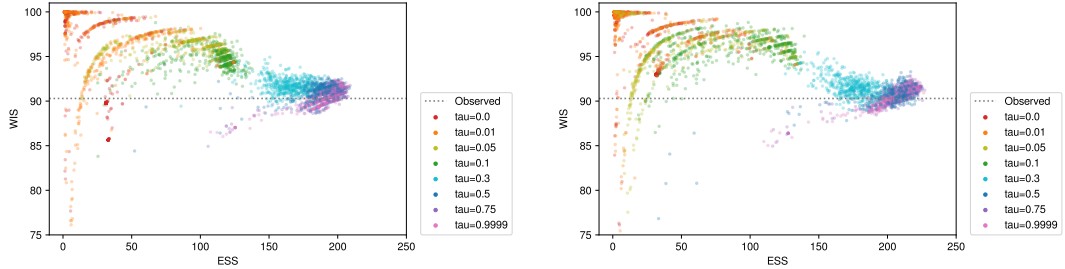

Figure 16: Validation performance (in terms of WIS and ESS) for all hyperparameter settings and all checkpoints considered during model selection. Left - baseline, Right - proposed.

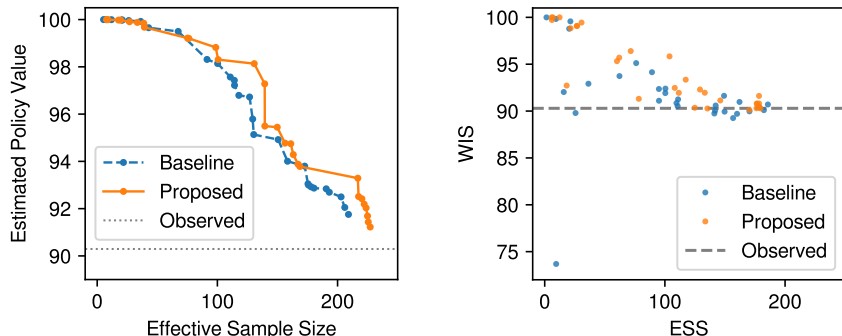

Figure 17: Left - Pareto frontiers of validation performance for the baseline and proposed approaches; Right - test performance of the candidate models that lie on the validation Pareto frontier. The validation performance largely reflects the test performance, and proposed approach outperforms the baseline in terms of test performance albeit with a bit more overlap.

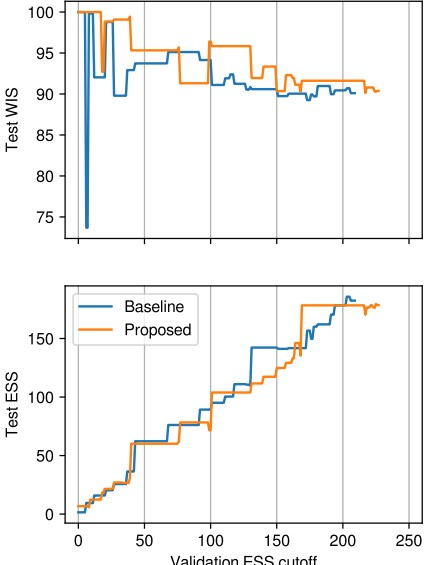

Figure 18: Model selection with different minimum ESS cutoffs. In the main paper we used ESS $\geq 200$; here we sweep this threshold and compare the resultant selected policies for both the baseline and proposed approach (only using candidate models that lie on the validation Pareto frontier). In general, across the ESS cutoffs, the proposed approach outperforms the baseline in terms of test set WIS value, with comparable or slightly lower ESS.