# OpenReview forum: "Leveraging Factored Action Spaces for Efficient Offline Reinforcement Learning in Healthcare"
_NeurIPS.cc/2022/Conference — NeurIPS 2022 Accept_

### Official Review · Reviewer_jzqP · 2022-07-11

**Rating:** 5
**Confidence:** 3
**Soundness:** 3 good
**Presentation:** 3 good
**Contribution:** 2 fair

**Summary:**

This paper proposes to decompose Q-function according to factorized action space. It conducts discussions on the bias-variance trade-off of the decomposed Q-function. Specifically, this paper provides sufficient conditions for zero bias and proves these conditions are not necessary. Furthermore, it shows that this form of Q-function decomposition leads to low variance because of the smaller lower bound on the empirical Rademacher complexity. Finally, it discusses what kinds of tasks can fulfill the sufficient conditions. The experiments on offline healthcare tasks demonstrate that the decomposed Q-function has the potential to help models improve their performance.

**Questions:**

The title of the article points out that this is an efficient offline RL approach and the sample efficiency is verified in the paper. However, I am still wondering whether the running time and computational cost of factored BCQ are significantly reduced compared to the baseline BCQ?
The comparison of the models in the experimental evaluation part may be insufficient. Whether the validity of linear decomposition of the Q-function is applicable to other models of offline RL, such as CQL[1], PQI[2], REM[3]?
In Figure 8a, why does factored BCQ almost never recommend IV fluids 1L-2L, which is inconsistent with clinician behavior policy?
[1] Kumar, A., Zhou, A., Tucker, G., & Levine, S. (2020). Conservative q-learning for offline reinforcement learning. Advances in Neural Information Processing Systems, 33, 1179-1191.
[2] Liu, Y., Swaminathan, A., Agarwal, A., & Brunskill, E. (2020). Provably good batch off-policy reinforcement learning without great exploration. Advances in neural information processing systems, 33, 1264-1274.
[3] Agarwal, R., Schuurmans, D., & Norouzi, M. (2020, November). An optimistic perspective on offline reinforcement learning. In International Conference on Machine Learning (pp. 104-114). PMLR.


**Limitations:**

The authors have discussed the limitations of their proposed approach in the manuscript, which may cause risky results in healthcare.

**Strengths And Weaknesses:**

Strengths: The paper is well-structured and easy to follow. This paper studies the guarantee of the unbiasedness of Q-function decomposition, which may inspire the researchers to focus on safe decomposition and improve performance in healthcare tasks. Besides, the proofs of bias-variance trade-off are provided in detail. On the sufficiency and necessity of these conditions, it validates them either by theoretical analysis or examples.
Weakness:
1. According to Figure 1, Q-function decomposition can help reduce some calculation steps in the model. However, this paper does not demonstrate the optimization of computational efficiency, especially when it comes to experiments.
2. According to Section 3.4, it seems that the sufficient conditions are very hard to satisfy which may lead the model rather impractical.
3. The lack of comparison to more algorithms makes the effectiveness of the decomposition somewhat less convincing.

---

> ### Author Response · Authors · 2022-08-02
> **Author Response to Reviewer jzqP**
>
> We thank the reviewer for their valuable feedback and suggestions. We address the weaknesses and questions below:
>
> **1. Computational efficiency**
>
> Thank you for raising this important point. Our current analysis focuses on statistical efficiency (variance) and its trade-off with approximation error (bias), but we agree computational efficiency is also an important aspect. Roughly speaking, to compute the values for all output heads in Fig 1, our approach has a time complexity of O(D) instead of O(exp(D)) of the baseline (measured in flops) and there is a clear saving of computational cost. Once the output heads are all computed, for the two common inference operations - $\operatorname{argmax}_a Q(s,a)$ and $\max_a Q(s,a)$ - the time complexity of baseline is also O(exp(D)). For the proposed approach, an optimized implementation has time complexity of O(D): we can perform argmax/max per sub-action and then concatenate/sum the results.
>
> In our current implementation, for simplicity, we did not implement the optimized version; instead, we make use of the subaction featurization matrix defined in Appendix B.4 (page 19; see code snippets in supplement `./RL_mimic_sepsis/4_BCQf/model.py`). While this makes autodiff easier to operate, it incurs additional computational steps relative to our analysis above. In particular, the forward pass includes a dense matrix multiplication with time complexity O(D exp(D)) flops, followed by an O(exp(D)) argmax/max operation, making it more expensive than the baseline. To help with implementation in settings where computational complexity might be a bottleneck, we will provide guidance on how to implement the optimized version of our approach.
>
> **2. Practicality of the proposed approach**
>
> In Sec 3.4 we outlined important practical considerations and illustrated when the theoretical conditions are (approximately) satisfied using several examples in the context of healthcare. When domain knowledge justifies the theoretical assumptions, we propose a simple method to capitalize on factored action spaces. On the other hand, we believe it’s equally important to understand when the proposed approach does *not* apply, and have explicitly discussed this in the paper (see L219). Additional examples where the assumptions are approximately satisfied: cooperative multi-agent games in finance where there’s a higher payoff if agents cooperate (positive interaction effects); intelligent tutoring systems that teaches basic arithmetic operations as well as fractions (which are distinct but related skills). We will include these examples in the camera-ready version.
>
> **3. Applicability to other algorithms**
>
> Yes, we believe our proposed linear decomposition is applicable to other algorithms. The key element of our proposed approach is the linearly decomposed Q-function induced by factored action spaces. This simple modification may be applied to any RL algorithm that has a value-based component, and we will emphasize this potential (currently only briefly mentioned on L14). However, we note that the theoretical guarantees may not translate exactly in every case. For example, CQL uses a lower bound Q-function definition that’s not based on the standard Bellman equation, so the unbiasedness conditions will need to be modified for CQL (our proof uses the Bellman equation for induction); this would be an interesting theoretical direction for future work.
>
> **4. MIMIC sepsis experiment, factored BCQ almost never recommend IV fluids 1L-2L (Fig 8)**
>
> Due to limited data, there will inevitably be errors in the learned Q-values, with which identifying the exact optimal action may be impractical. From our data, of the 1913 states that correspond to the region in the red box where factored BCQ recommends large doses of IV fluids > 500mL, we found that: while actions with 1L-2L fluids are optimal for only 10 states, they are considered the second/third best option for 787 states, and in all 1913 cases they are considered better than no fluids. Given the challenging nature of identifying the best treatments from offline data, we expect our proposed approach to be combined with other offline RL techniques that do not aim to identify the single best action (e.g., learning dead-ends [[Fatemi et al. 2021](https://openreview.net/forum?id=4CRpaV4pYp)] or set-valued policies [[Tang et al. 2020](http://proceedings.mlr.press/v119/tang20c.html)])

---

> > ### Comment · Reviewer_jzqP · 2022-08-08
> > **Thanks for your responses**
> >
> > Thanks to the authors for their responses. I do feel the paper has improved slightly with the edits, while it seems still well described as "Technically solid paper where reasons to accept outweigh reasons to reject".

---

### Official Review · Reviewer_yW3d · 2022-07-11

**Rating:** 7
**Confidence:** 3
**Soundness:** 3 good
**Presentation:** 3 good
**Contribution:** 3 good

**Summary:**

This study improved and extended the standard reinforcement learning (RL) to factored action space. A form of linear Q-function decomposition was proposed to handle factored action space. The novel method has been analyzed from a theoretical perspective, where the authors discussed the sufficient and necessary conditions for unbiases and studied its effect on variance reduction. The authors also demonstrated the proposed method through empirical experiments based on a simulation study and real-world MIMIC data analysis. The novel method provides both theoretical insights and empirical evidence for RL practitioners to consider this simple linear decomposition approach to factored action space.

**Questions:**

1.	Based on experimental evaluation (Figure 7), I was unable to observe significant improvement (in terms of clinical, real-world applicability) when compared with baseline BCQ and factored BCQ (90.44±2.44 vs 91.62±2.12).Therefore, it might be hard to conclude that the proposed factored BCQ outperforms the baseline.

2.	Based on Figure 6, the authors also claim better performance for a small sample size. I have concerns on this claim due to the lack of sufficient evidence. The variation is large for a small sample size, and the selection of samples would affect the result. Also, I couldn’t find the authors’ discussions on why the proposed method would get higher performance for small sample size, especially from the theoretical perspective.

3.	The authors state that domain knowledge may be used to inform the applicability of the approach in practice. However, very little information in the main text address on how domain knowledge can be involved. This is an important question for real-world applications, thus would like to suggest the authors to provide more details for justifications.

4.	If the authors claim that the method works well only for a limited sample size, I would suggest the authors do some analyses to determine the sample size cut-off (or give a quantitative/analytical formula to calculate). When the sample size is large enough, the empirical research shows that the proposed method is less useful and effective as compared to the baseline. Therefore, knowing the sample size cut-off is essential.

5.	If possible, adding another real-world use case would be ideal to justify the generalizability and applicability of the proposed method.


**Limitations:**

1.	The performance improvement brought by the proposed method seems marginal. I suggest the authors add some other performance indicators and provide additional discussions.

2.	There is a lack of interpretability for the proposed method. The authors are suggested to elaborate on how to interpret the results at the individual level and how this can be applied to clinical practice.

3.	There are some preconditions required for using this proposed method. I would suggest the author provide some discussions and give more guidelines for future researchers who would like to use your method.

4.	Please also add more on the importance of your method and how your method can address the unmet need of current literature.

**Strengths And Weaknesses:**

Strength: The research question is novel, and the authors found the unmet need of factored action spaces in the RL approach. Solving this problem will facilitate the use of offline RL in a broader application, especially in healthcare. The methodology is well-explained with details. This paper is complete with theoretical insights and empirical experiments.

Weakness: The proposed method showed marginal improvement compared to the existing method. I also have some conservations on the applicability of the method (where many preconditions are required). The improvement has not addressed the key limitations of offline RL (i.e., how to improve exploration, discover high-reward regions and solve distributional shift problems).

---

> ### Author Response · Authors · 2022-08-02
> **Author Response to Reviewer yW3d**
>
> We thank the reviewer for their valuable feedback and suggestions. Below, we respond to the questions and weaknesses raised in this review:
>
> **1. Key challenges of offline RL**
>
> Our contribution is complementary to popular offline RL approaches and goes beyond pessimism only methods by leveraging domain knowledge. When domain knowledge is present such as for healthcare settings, we argue that one should make good use of it. Our work demonstrates one way to do so, and our proposed linear Q decomposition can be combined with new solutions tackling the other challenges of offline RL (e.g. distribution shift).
>
> **2. MIMIC experiment (Fig 7): performance improvements seem marginal, interpretability, and applied to clinical practice**
>
> We opted to conduct empirical experiments in challenging realistic settings rather than creating contrived examples where our approach works perfectly (which is largely captured by the theoretical analysis). Quantitatively, our proposed approach leads to a small but consistent improvement over the baseline: since this is a high-stakes domain, it is unlikely an RL policy will lead to drastically better survival rate. For qualitative evaluation, we worked closely with domain experts on the problem formulation and interpretation of results, and found our approach learns more sensible policies (Fig 8) by leveraging the overlap among treatment combinations. Combined, we have some initial confidence that our approach learns better policies, but with regard to real-world applicability, our experiments serve more as a proof-of-concept. However, we *cannot stress enough* that further investigations, such as interpreting policies at individual level, are essential before such RL algorithms are deployed in practice. While our current work is not focused on interpretable methods, we plan to incorporate some of the recent techniques in interpretable RL such as contrastive explanations (e.g. https://arxiv.org/abs/2207.06269).
>
> **3. Fig 6: does proposed method have better performance and why**
>
> Following best practices (e.g. https://openreview.net/forum?id=uqv8-U4lKBe), to lessen the effect of randomness, our reported results are not from a single run; as mentioned on L255, we repeated each setting for 10 runs with different random seeds and plotted the median performance with IQR as well as individual points. While the variation is large, we observe a consistent trend. As for “why” performance is better for ​​small sample sizes: on L264 we explained “This is because variance decreases with sample size but the bias incurred by the factored approximation does not change. Once there are enough samples, reductions in variance are no longer advantageous and the incurred bias dominates the performance.” We will clarify the connection between this explanation and the theoretical analysis on bias-variance trade-off (Sec 3.3.1 on page 5).
>
> **4. Small sample size cutoff**
>
> This depends on the amount of bias. We plan to extend the finite-sample results in https://proceedings.mlr.press/v97/chen19e.html (Appendix C) of the Bias + Variance breakdown and derive the "break-even" point given a specific level of bias. It should be noted that in many of the practical scenarios in healthcare, we have access to very limited samples and strong domain knowledge, and we expect our approach to be appropriate and lead to performance improvements for many tasks.
>
> **5. Guidelines for applicability, domain knowledge** (see also: overall comment)
>
> In Sec 3.4, we outlined important practical considerations and illustrated when the theoretical conditions are (approximately) satisfied using several examples in the context of healthcare. When domain knowledge justifies the theoretical assumptions, we propose a simple method to capitalize on factored action spaces. On the other hand, we believe it’s equally important to understand when the proposed approach does *not* apply, and have explicitly discussed this in the paper (see L219 where we present examples of adverse drug interactions).
>
> Our work focuses on healthcare because we are most familiar with this setting and have expertise in healthcare. We worked closely with clinician(s) to guide the RL problem formulation and qualitative result interpretation. At present, we do not have plans to investigate other application areas, but if we are allowed to speculate, we think our proposed approach would work well in settings where domain knowledge is available and can help justify our theoretical conditions, e.g., cooperative multi-agent games in finance where there’s a higher payoff if agents cooperate (positive interaction effects); intelligent tutoring systems that teach basic arithmetic operations as well as fractions (which are distinct but related skills). We will include these examples in the camera-ready version. We hope our work serves as good inspiration for researchers working in other application domains of (offline) RL that exhibit similar structures.

---

> > ### Comment · Reviewer_yW3d · 2022-08-10
> > **Thanks for your response**
> >
> > I believe the manuscript has been improved and now is more sound. Though I still have some reservations, this study seems to bring in new contributions to the field. Thus, I increased my recommendation score.

---

### Official Review · Reviewer_CUKv · 2022-07-11

**Rating:** 6
**Confidence:** 4
**Soundness:** 3 good
**Presentation:** 2 fair
**Contribution:** 3 good

**Summary:**

This paper proposes the factored action spaces for offline RL in the applications of healthcare AI. The authors aim to leverage the factored action space in the form of linear Q-function decomposition to improve the sample efficiency compared with the baselines with combinational action spaces. The theoretical guarantees show that the linear Q-function decomposition can lead to zero bias under mild conditions. And even if the assumptions are violated, the method can still achieve good performances (nearly policy optimality). Empirical analysis verifies the proposed claims and effectiveness of the approach on simulated and real-world healthcare RL benchmarks.

**Questions:**

#### **1. About the autoregressive factorization**

Can authors give some analysis on both theoretical and empirical views on the scenarios where the actions are entirely in the autoregressive factorization? I notice that the theory part can be extended to this, and the empirical evaluation includes cases where a few actions are not independent. However, the entire autoregressive factorization, which is very common in RL with sub-actions, is not analyzed in the current version. In autoregressive factorization, the state would be augmented with the previously executed actions. Any discussion or analysis would be highly appreciated.

#### **2. About the state space**

In the current version, each sub-action is based on the full state space. I think that why not leverage the local factorization or factored state space, where the state-action space is entirely factorized, and the state space of each sub-action would be much smaller. I think the sample efficiency would also be improved with the factored state-(sub)action space. **Please note that this is only a discussion. There is no need to run any experiment on this during the rebuttal phase.**

 ### **References**

[1] Pitis, Silviu, Elliot Creager, and Animesh Garg. "Counterfactual data augmentation using locally factored dynamics." Advances in Neural Information Processing Systems 33 (2020): 3976-3990.

[2] Huang, Biwei, et al. "Action-sufficient state representation learning for control with structural constraints." ICML 2022.


**Limitations:**

The limitations are more related to the algorithmic contribution and empirical evaluation (listed in the weakness section). I will increase my score if the authors give justifications during the rebuttal phase.

**Strengths And Weaknesses:**

## *Strengths*

#### **1. Technical soundness and significance**

The paper proposes the factored action space, where the action space is expressed as a Cartesian product of a few sub-actions spaces. Though similar ideas and methods have been proposed in the past few years, this paper gives a detailed and comprehensive analysis in both theoretical and empirical manners. The theory parts are complete and technically sound, and evaluation with practical consideration also verifies the theoretical claims. With the analyzed theorems in this paper, factored approaches (either in state-action space or action space) in RL would be more broadly applicable.

#### **2. Presentation**

Though the presentation can be further improved for better readability (see the weaknesses points below), the overall logical flow is clear, and the concrete examples (Fig. 2-Fig. 3) are helpful for readers to understand the theorems.

## *Weakness*

#### **1. The contributions on the algorithmic aspects are vague**

As the authors mentioned in the related work section, several works have been proposed to explore the benefits of factored action space for both model-based and model-free RL. Yes, this work offers the solution for leveraging factored actions with value-based methods in
offline RL. However, from the current version, I cannot tell the significant contributions of the algorithmic aspects of learning the factored space. The Cartesian product for action space has been used extensively in previous works (i.e., [1]). It would be better to explicitly list the contributions to the algorithmic aspects in the revised version. A table with comparisons of all related approaches would be helpful.

#### **2. About evaluation**

More complicated benchmarks (e.g., Mujoco control tasks, DOTA2, StarCraft, etc.) have been tested in other works using factored action spaces. Can this framework also be applicable for other commonly-used RL benchmarks? **Please note that running on these benchmarks during the rebuttal phase is not a must, but any discussion or analysis would be highly appreciated.**

#### **3. Possible directions to improve the writing for better readability**

#### -> 3.1 Giving algorithmic frameworks into the main paper

The authors can consider adding the algorithmic frameworks as algorithm pseudo-code or figures. The framework can explain the pipelines of learning or exploiting the factored spaces. Moving some justification contents in Section 3.3.2 into the appendix can save room for this (since I feel like the propositions in 3.3.2 is already very clear).

#### -> 3.2 Adding one background section on factored MDP and factored action space

The related work section in the appendix briefly introduces factored MDP and action space in RL. I think it is better to briefly give some formal definitions in the background or preliminary sections in the main paper.


### **References**

[1] PIERROT, Thomas, et al. "Factored Action Spaces in Deep Reinforcement Learning." (2020).

---

> ### Author Response · Authors · 2022-08-02
> **Author Response to Reviewer CUKv**
>
> We thank the reviewer for their valuable feedback and suggestions. We are encouraged that the reviewer appreciated the technical soundness/significance and presentation of our work. We address the weaknesses and questions below:
>
> **1. Clarifying algorithmic contributions**
>
> In the related work section (Appx A.2), we have discussed how our work relates to and differs from prior literature related to action space factorizations on factored MDP and single/multi-agent RL. We moved the related work to the appendix to save space, but realize this was a mistake. We will move this section back to the main paper prior to publication. We have also created the following table of comparisons. In particular, most prior work assumes either explicit state space factorization or known state space abstractions (e.g. each agent has its own observations in the multi-agent setting), whereas our main theoretical result only requires implicit abstractions to exist. While similar Q decomposition was used in Tavakoli et al. 2018, we are the first to present theoretical guarantees and bias-variance analysis for this setting. Lastly, our paper focuses on value-based methods since those have seen the most success in offline RL; it would be interesting to extend our results to policy-based and model-based settings.
>
> |      |  Policy-based?  | Model-based? | Value-based? | Linear value decomposition? | Known state factorization or abstraction? | Unbiasedness guarantees? |
> | --- | --- | --- | --- | --- | --- | --- |
> | Osband & Van Roy 2014;  Lu et al. 2021 |  | ✔ |  |  | ✔ |  |
> | Sallans & Hinton 2004 | ✔ |  |  |  | ✔ |  |
> | Pierrot et al. 2021; Spooner et al. 2021 | ✔ |  |  |  | ✗ |  |
> | Koller & Par 1999; Guestrin et al. 2003; Strehl et al. 2007; Delgado et al. 2011 |  |  | ✔ (V) | ✔ | ✔ |  |
> | Sharma et al. 2017 | ✔ |  | ✔ (Q) | ✔+¹ | ✔ |  |
> | VDN, Sunehag et al. 2017 |  |  | ✔ (Q) | ✔ | ✔ |  |
> | QMIX, Rashid et al. 2018 |  |  | ✔ (Q) | *² | ✔ |  |
> | BDQ, Tavakoli et al. 2018 |  |  | ✔ (Q) | *² | ✗ |  |
> | → **This work** |  |  | ✔ (Q) | ✔ | ✗ | ✔ |
>
> ¹ Empirically tested various “combination” functions including linear.
> ² Both QMIX and BDQ do not aggregate the sub-Q functions; instead, they aggregate the argmax sub-actions.
>
>
> **2. Extending empirical evaluations** (see also: cdHc)
>
> Our work focuses on healthcare because we are most familiar with this setting and have expertise in healthcare. We worked closely with clinician(s) to guide the RL problem formulation and qualitative result interpretation. At present, we do not have plans to investigate other application areas, but if we are allowed to speculate, we think our proposed approach would work well in settings where domain knowledge is available and can help justify our theoretical conditions, e.g., multi-agent games in finance where there’s a higher payoff if agents cooperate (positive interaction effects); intelligent tutoring systems that teaches basic arithmetic operations as well as fractions (which are distinct but related skills). We hope our work serves as good inspiration for researchers working in other domains that use RL that exhibit similar problem structures.
>
> **3. Autoregressive action factorization**
>
> Thank you for pointing this out. Our work focuses on the setting of *independent* action space factorization, and we did not consider *autoregressive* factorization (cf. https://openreview.net/forum?id=naSAkn2Xo46); we’ll make sure to state this explicitly in the camera-ready version. We agree with you that Thm 1 can be extended to the autoregressive setting by augmenting states with previously executed actions, for example, Eqn (2) would become $p(s’|s,a) = \prod_{d=1}^{D} p_d(z_d’|z_d, a_1, \cdots, a_d)$. We believe autoregressive factorizations of action space are relevant for healthcare as well (e.g., an action corresponds to first deciding whether a drug should be used (binary 0/1), and then selecting dosage if the drug is used). We think this is an exciting future direction and plan to pursue it.
>
> **4. Using factored *state* space**
>
> We agree that when additional knowledge about the state space is available, then it should be leveraged to make learning more efficient. E.g., for a tabular problem with factored state space with $\otimes S_d$ and factored action space $\otimes A_d$, the number of free parameters in the “factored Q-table” is $\sum |S_d||A_d|-D+1$ instead of $(\prod|S_d|)(\prod|A_d|)$ of the full Q-table (cf. L149). Since oftentimes in healthcare the state space factorization is unknown, we focus on action space factorization. E.g., in the MIMIC sepsis domain, it is not trivial to define the factorization (or the abstraction) of the raw physiological feature space for vasopressors and IV fluids. Our theoretical analysis shows that the proposed approach only requires implicit state space factorizations and is able to handle such settings, but we will mention potential extensions in the camera-ready version.

---

> > ### Comment · Reviewer_CUKv · 2022-08-05
> > **Feedback**
> >
> > Thanks for the detailed and thoughtful response. The rebuttal addresses my major concerns.

---

### Official Review · Reviewer_cdHc · 2022-07-12

**Rating:** 7
**Confidence:** 3
**Soundness:** 4 excellent
**Presentation:** 4 excellent
**Contribution:** 3 good

**Summary:**

This paper considers the use of a factored Q-function representation for reinforcement learning that takes advantage of a factored action space in certain "appropriate" MDPs, where "appropriate" is characterized both theoretically and qualitatively/practically in the paper. The authors show empirically that their factored representation can help in low-data healthcare settings, even when it is not a perfect match for the environment. It is argued that this is because it allows for a better bias-variance trade-off and provides a structural basis to generalize outside of the empirical distribution.

**Questions:**

A useful tactic in cases where we have an "approximately correct" structure, as you have in your experiments is to use a residual (even LSTMs / Resnets are examples of this, where identity is the approximately correct structure). So here, you could have $Q^\pi(s, \textbf{a}) = \sum q_d(s, a_d) + \mathcal{R}(s, a)$, where the residual function $\mathcal{R}$ can capture interactions between the components (and can be heavily regularized in cases of low data). It may be interesting for you to consider this approach in your applications. Not really a question, I know.

I guess my most relevant question is: have you looked at the FMDP literature at all? I don't see it mentioned anywhere in the paper. I wonder if your Theorem can be related to some of their representation theorems by a simple transform of the state space (that entangles the previously disentangled state features). Could be interesting / worth looking into.

**Limitations:**

I think this paper adequately addressed the limitations and potential negative societal impact of their work. The authors discuss the stringent conditions of their representations at length, and their experiments are performed in environments that *do not* satisfy their sufficient conditions.

**Strengths And Weaknesses:**

This is a good paper. At first I was put off by the specificity of the representation---it really did seem to me that the factored action space representation used here was too specific to be of any practical use, but I felt that the authors did well to combat this and demonstrated applicability in healthcare at the very least. The presentation is clear and well done, and the development is logically consistent and flows well. While I found many of the theoretical bits "obvious" (e.g., props 2-4), I also felt that their presence adds to the overall depth of the paper as more or less a complete treatment of this "factored action space" representation of the Q-function. The fact that the abstraction used in Theorem 1 can be implicit is what makes this Theorem, and the paper as a whole, interesting, and sufficiently differentiated from, say, the literature on factored MDPs (but I'm not an expert in FMDPs). The empirical portions are well done---experiments are highly relevant, and the connection between the experiments and the method is plain---and the discussion is insightful.

I have seen, but not read in any detail, many works in multi-agent RL (and otherwise), which use factored action spaces (there are 69 results for "factored action spaces" on Scholar). I am trusting the author's lit review on the novelty point, and I hope that at least one of the other reviewers is sufficiently familiar with the literature to give a good opinion on novelty. To me, it seems sufficiently novel, but I lack broad knowledge of other work on factored action spaces.

I'm going to give this paper an "Accept" rating, since I don't see easy ways to improve it, and it feels complete. That being said, I do think the description for "Weak Accept" (which is pretty strong?) is more accurate, insofar as this paper is rather specific.

---

> ### Author Response · Authors · 2022-08-02
> **Author Response to Reviewer cdHc**
>
> We thank the reviewer for their valuable feedback and suggestions. We are encouraged that the reviewer found our paper to be clearly written and appreciated the novelty and relevance of our work. We address the main topics of discussion below:
>
> **1. Specificity of the application area** (see also: overall comment, CUKv)
>
> Our work focuses on healthcare because we are most familiar with this setting and have expertise in healthcare. We worked closely with clinician(s) to guide the RL problem formulation and qualitative result interpretation. At present, we do not have plans to investigate other application areas, but if we are allowed to speculate, we think our proposed approach would work well in settings where domain knowledge is available and can help justify our theoretical conditions, e.g., cooperative multi-agent games in finance where there’s a higher payoff if agents cooperate (positive interaction effects); intelligent tutoring systems that teaches basic arithmetic operations as well as fractions (which are distinct but related skills). We hope our work serves as good inspiration for researchers working in other application domains of (offline) RL that exhibit similar problem structures.
>
> **2. Modeling “residual” interactions and regularization**
>
> Thank you for this suggestion. While we did not formally consider this setting, we mentioned on L223 that “one can either explicitly encode the interaction terms or resort back to a combinatorial action space” when the interaction effect is not negligible and can lead to suboptimal performance. It’s possible to extend our theoretical analyses to this setting (e.g., a sufficient condition is to modify the Eqn (3) reward condition to account for the interaction while keeping the Eqn (2) transition and Eqn (4) policy the same -- this allows for a wider class of permissible reward functions); we plan to outline this setting in the appendix.
>
> Your suggestion on regularizing the residual terms is an interesting idea that we did not think about; we believe it represents a good heuristic that improves the practicality of our approach, and also points to areas for theoretical research which we are excited to investigate in our future work. We also believe that your suggestion combined with [Tavakoli et al. 2021](https://openreview.net/forum?id=Xv_s64FiXTv) (which allows one to systematically consider interactions of certain “ranks” e.g., limiting it to only two-way or three-way interactions), would be a nice extension of our current work.
>
> **3. Relation to Factored MDP** (see also: overall comment, CUKv)
>
> Yes, we have carefully studied the FMDP literature. We mentioned FMDP on L63 in Sec 2.2 and also related work in Appx A.2. We moved the related work to the appendix to save space, but realize this was a mistake. We will move this section back to the main paper prior to publication. Our work focuses on factored action spaces and differs from the majority of the FMDP literature that concerns state space factorizations. You are exactly right that our main result in Thm 1 can be seen as an extension of some related FMDP results on factored value functions (which we state in Proposition 5 in Appendix with a focus on $Q$-functions instead of state value functions $V$; cf. Koller & Parr, “Computing factored value functions for policies in structured MDPs”, IJCAI-99), achieved via a transformation of the state space (i.e. “state abstractions” in Theorem 1), but importantly our analysis shows that this transformation does not need to be known for the linear decomposition to exist.

---

### Author Response · Authors · 2022-08-02
**Summary of Responses**

We thank the reviewers for their valuable feedback! Reviewers found the paper to be well-written, with novel theoretical insights and highly relevant experiments. We have responded to the questions/comments in each review below, but here we want to highlight some of the common themes raised by multiple reviewers.

**Algorithmic contributions & related work (cdHc, CUKv)**

In our submission, we included the related work section in the appendix (page 15) due to space constraints. We realize this is a mistake, so in the camera-ready, we plan to move this section to the main paper. In the related work section, we have discussed how our work relates to and differs from representative works in FMDP, multi-agent RL, and previous approaches using factored actions. Below, we further outline the differences of this work compared to previous literature. We will add this to the camera-ready version of the paper.


|      |  Policy-based?  | Model-based? | Value-based? | Linear value decomposition? | Known state factorization or abstraction? | Unbiasedness guarantees? |
| --- | --- | --- | --- | --- | --- | --- |
| Osband & Van Roy 2014;  Lu et al. 2021 |  | ✔ |  |  | ✔ |  |
| Sallans & Hinton 2004 | ✔ |  |  |  | ✔ |  |
| Pierrot et al. 2021; Spooner et al. 2021 | ✔ |  |  |  | ✗ |  |
| Koller & Par 1999; Guestrin et al. 2003; Strehl et al. 2007; Delgado et al. 2011 |  |  | ✔ (V) | ✔ | ✔ |  |
| Sharma et al. 2017 | ✔ |  | ✔ (Q) | ✔+¹ | ✔ |  |
| VDN, Sunehag et al. 2017 |  |  | ✔ (Q) | ✔ | ✔ |  |
| QMIX, Rashid et al. 2018 |  |  | ✔ (Q) | *² | ✔ |  |
| BDQ, Tavakoli et al. 2018 |  |  | ✔ (Q) | *² | ✗ |  |
| → **This work** |  |  | ✔ (Q) | ✔ | ✗ | ✔ |

¹ Empirically tested various “combination” functions including linear.
² Both QMIX and BDQ do not aggregate the sub-Q functions; instead, they aggregate the argmax sub-actions.


**Domain knowledge, practical guidelines, & real-world use cases (cdHc, CUKv, yW3d, jzqP)**

In Sec 3.4, we outlined important practical considerations and illustrated when the theoretical conditions are (approximately) satisfied using several examples in the context of healthcare. When domain knowledge justifies the theoretical assumptions, we propose a simple method to capitalize on factored action spaces. On the other hand, we believe it’s equally important to understand when the proposed approach does *not* apply, and have explicitly discussed this in the paper (see L219 where we present examples of adverse drug interactions).

Our work focuses on healthcare because we are most familiar with this setting and have expertise in healthcare. We worked closely with clinician(s) to guide the RL problem formulation and qualitative result interpretation. At present, we do not have plans to investigate other application areas, but if we are allowed to speculate, we think our proposed approach would work well in settings where domain knowledge is available and can help justify our theoretical conditions, e.g., cooperative multi-agent games in finance where there’s a higher payoff if agents cooperate (positive interaction effects); intelligent tutoring systems that teach basic arithmetic operations as well as fractions (which are distinct but related skills). We will include these examples in the camera-ready version. We hope our work serves as good inspiration for researchers working in other application domains of (offline) RL that exhibit similar structures.

---

### Meta-Review · Area_Chair_bSwk · 2022-08-27

**Recommendation:** Accept
**Confidence:** Certain

**Metareview:**

Reviewers agree that the problem of factored action spaces in RL is important and that this paper makes novel contributions to this setting.  The reviewers were satisfied with the post-rebuttal discusion and have converged on an accept recommendation.

On revision, the reviewers request that the authors revise the paper according to the clarifications that occurred during post-rebuttal discussion.

Also, for context, it's important to note that the concept of factored action spaces goes back a long way in the factored MDP literature and I would request the authors to acknowledge this in their related work discussion as they prepare their final revision.  To the best of my knowledge, the first mention of factored action spaces is in a 1996 multiagent MDP paper:

Craig Boutilier.  Planning, Learning and Coordination in Multiagent Decision Processes. (1996)
https://www.cs.toronto.edu/~cebly/Papers/tark96.pdf

Somewhat more recently, the following paper presented a sequential hindsight method for compositional MDPs that is an upper bound approximation for (weakly) coupled MDPs.  I mention this specific paper since it discusses theoretical results relating to factored action MDP approximations and also presents a simple approximate decomposition methodology that I have found hard to beat empirically:

Aswin Raghavan, Saket Joshi, Alan Fern, Prasad Tadepallia, Roni Khardon.  Planning in Factored Action Spaces with Symbolic Dynamic Programming. (2012)
https://ojs.aaai.org/index.php/AAAI/article/view/8364


**Award:**

No

---

### Decision · Program_Chairs · 2022-09-14

Accept